

# Faunal carbon flows in the abyssal plain food web of the Peru Basin have not recovered during 26 years from an experimental sediment disturbance

Tanja Stratmann[1]*, Lidia Lins[2]†, Autun Purser[3], Yann Marcon[3]‡, Clara F. Rodrigues[4], Ascensão Ravara[4], Marina R. Cunha[4], Erik Simon-Lledó[5], Daniel O. B. Jones[5], Andrew K. Sweetman[6], Kevin Köser[7], Dick van Oevelen[1]

[1]NIOZ Royal Netherlands Institute for Sea Research, Department of Estuarine and Delta Systems, and Utrecht University, P.O. Box 140, 4400 AC Yerseke, The Netherlands.
[2]Marine Biology Research Group, Ghent University, Krijgslaan 281 S8, 9000 Ghent, Belgium.
†present address: Senckenberg Research Institute, Senckenberganlage 25, 60325 Frankfurt am Main, Germany
[3]Deep Sea Ecology and Technology, Alfred Wegener Institute, Am Handelshafen 12, 27570 Bremerhaven, Germany.
‡MARUM-Center for Marine Environmental Sciences, General Geology - Marine Geology, University of Bremen, D-28359 Bremen, Germany
[4]Departamento de Biologia & Centro de Estudos do Ambiente e do Mar (CESAM), Departamento de Biologia, Universidade de Aveiro, Campus de Santiago, 3810-193 Aveiro, Portugal
[5]National Oceanography Centre, University of Southampton Waterfront Campus, European Way, Southampton SO14 3ZH, UK.
[6]Marine Benthic Ecology, Biogeochemistry and In-situ Technology Research Group, The Lyell Centre for Earth and Marine Science and Technology, Heriot-Watt University, Edinburgh EH14 4AS, UK.
[7]GEOMAR Helmholtz Centre for Ocean Research, FE Marine Geosystems, Wischhofstr 1-3, 24148 Kiel, Germany.

*Correspondence to*: Tanja Stratmann (tanja.stratmann@nioz.nl)

## Abstract

Future deep-sea mining for polymetallic nodules in abyssal plains will impact the benthic ecosystem, but it is largely unclear whether this ecosystem will be able to recover from mining disturbance and if so, at what time scale and to which extent. In 1989, during the 'DISturbance and reCOLonization' (DISCOL) experiment, a total of 22% of the surface within a 10.8 km² large circular area of the nodule-rich seafloor in the Peru Basin (SE Pacific) was ploughed to bury nodules and mix the surface sediment. This area was revisited 0.1, 0.5, 3, 7, and 26 years after the disturbance to assess macrofauna, megafauna and fish density and diversity. We used this unique abyssal faunal time series to develop carbon-based food web models for disturbed (sediment inside the plough tracks) and undisturbed (sediment inside the experimental area, but outside the plough tracks) sites. We developed a linear inverse model (LIM) to resolve carbon flows between 7 different feeding types within macrofauna, megafauna and fish. The total faunal biomass was always higher at the undisturbed sites compared to the disturbed sites and 26 years post-disturbance the biomass at the disturbed sites was only 54% of the biomass at undisturbed sites. Fish and sub-surface deposit feeders experienced a particularly large temporal variability in biomass and model-reconstructed respiration rates making it difficult to determine disturbance impacts. Deposit feeders were least affected by the disturbance, with respiration, external predation and excretion levels only reduced by 2.6% in the sediments disturbed 26-years ago compared



with undisturbed areas. In contrast, the respiration rate of filter and suspension feeders was still 79.5% lower after 26 years when comparing the same sites. The 'total system throughput' ($T_{..}$), i.e. the total sum of carbon flows in the food web, was always higher at undisturbed sites compared to the corresponding disturbed sites and was lowest at disturbed sites directly after the disturbance ($8.63 \times 10^{-3} \pm 1.58 \times 10^{-5}$ mmol C m$^{-2}$ d$^{-1}$). Therefore, 26 years after the DISCOL disturbance, the

throughput discrepancy between the undisturbed and the disturbed sediment was still 56%. From these results we conclude that C cycling within the fauna compartments of an abyssal plain ecosystem remains reduced 26 years after physical disturbance, and that a longer period of time is required for the system to recover from such a simulated small scale deep-sea mining experimental disturbance.

## 1 Introduction

Abyssal plains cover approximately 50% of the world's surface and 75% of the seafloor (Ramirez-Llodra et al., 2010). The abyssal seafloor is primarily composed of soft sediments consisting of fine-grained erosional detritus and biogenic particles (Smith et al., 2008). Occasionally, hard substrate occurs occasionally in the form of clinker from steam ships, glacial drop stones, outcrops of basaltic rock, whale carcasses, and marine litter (Amon et al., 2017; Kidd and Huggett, 1981; Radziejewska, 2014; Ramirez-Llodra et al., 2011; Ruhl et al., 2008). In some soft sediment regions, islands of hard substrate are provided by

polymetallic nodules, authigenically formed deposits of metals, which grow at approximate rates of 2 to 20 mm per million years (Guichard et al., 1978; Kuhn et al., 2017). These nodules have the shape and size of cauliflower, cannon balls or potatoes, and are found on the sediment surface and in the sediment at depths between 4000 and 6000 m in areas of the Pacific, Atlantic and Indian Ocean (Devey et al., 2018; Kuhn et al., 2017).

Polymetallic nodules are rich in metals, such as nickel, copper, cobalt, molybdenum, zirconium, lithium, yttrium and rare earth elements (Hein et al., 2013), and occur in sufficient densities for potential exploitation by the commercial mining industry in the Clarion-Clipperton Fracture Zone (CCFZ; equatorial Pacific), around the Cook Islands (equatorial Pacific), in the Peru Basin (E Pacific) and in the Central Indian Ocean Basin (Kuhn et al., 2017). Extracting these polymetallic nodules during deep-sea mining operations will have severe impacts on the benthic ecosystem, such as the removal of hard substrate (i.e.

nodules) and the food-rich surface sediments from the seafloor, physically causing the mortality of organisms within the mining tracks and re-settlement of resuspended particles (Levin et al., 2016; Thiel and Tiefsee-Umweltschutz, 2001). Defining regulations on deep-sea mining requires knowledge on ecosystem recovery from these activities, but to date information on these rates is not extensive (Gollner et al., 2017; Jones et al., 2017; Stratmann et al., 2018; Stratmann et al., in review; Vanreusel et al., 2016). Especially the recovery of ecosystem functions, such as food web structure and carbon (C) cycling, from deep-

sea mining is understudied.




In the Peru Basin (SE Pacific), small-scale deep-sea mining activities were simulated during the 'DISturbance and reCOLonization' experiment (DISCOL) in 1989. A 10.8 km$^2$ large circular area was ploughed diametrically 78 times with a 8 m-wide plough-harrow to bury the surface nodules into the sediment (Thiel and Schriever, 1989). This experimental disturbance resulted in a heavily disturbed centre and a less affected periphery of the DISCOL area (Bluhm, 2001; Foell et al., 1990; Foell et al., 1992). Over 26 years the region was re-visited five times to assess the Post-Disturbance (PD) situation: directly after the disturbance event, March 1989: (hereafter referred to as 'PD$_{0.1}$'); half a year later, September 1989: 'PD$_{0.5}$'; three years later, January 1992: 'PD$_3$'; seven years later, February 1996: 'PD$_7$'; 26 years later, September 2015: 'PD$_{26}$'. Following the original definition by Bluhm (2001), we denote sites within the DEA (DISCOL Experimental Area), but not directly disturbed by the plough harrow as 'undisturbed sites' and sites that were directly impacted by the plough harrow as 'disturbed sites' (Bluhm, 2001). During subsequent visits, densities of macrofauna and megafauna were assessed, but data on meiofauna and microbial communities were only sparsely collected. Therefore, the food web models presented in this work cover a period of 1989 to 2015 and contain macrofauna, megafauna and fish.

Linear inverse modelling (LIM) is an approach that has been developed to disentangle carbon flows between food web compartments for data-sparse systems (Klepper and Van de Kamer, 1987; Vézina and Platt, 1988). It has been applied to assess differences in C and nitrogen (N) cycling in various ecosystems, including the abyssal plain food web at Station M (NE Pacific) under various particulate organic carbon (POC) flux regimes (Dunlop et al., 2016), and a comparison of food web flows between abyssal hills and plains at the Porcupine Abyssal Plain (PAP) in the north-eastern Atlantic (Durden et al., 2017). LIM is based on the principle of mass balancing various data sources (Vézina and Platt, 1988), i.e. faunal biomasses and physiological constraints, that are implemented in the model, either as equality or inequality equations, and these are solved simultaneously (van Oevelen et al., 2010). A food web model almost always includes more inequalities than equalities, i.e. it is mathematically under-determined, which implies that an infinite number of solutions will solve the models. In this case, a likelihood approach can be used to generate a large dataset of possible solutions for the model (van Oevelen et al., 2010), from which the mean and standard deviations for each flow is calculated. Food web models from different sites and/or points in time can be compared quantitatively by calculating network indices, such as the 'total system throughput' ($T..$) that sums all carbon flows in the food web (Kones et al., 2009). Hence, a decrease in the difference of $T..$ between the food webs from undisturbed and corresponding disturbed sites ($\Delta T..$) over time is taken as a sign of ecosystem recovery following disturbance.

In this study, benthic food-web models were developed for undisturbed sites and disturbed sites at DISCOL to assess whether faunal biomass and trophic composition of the food webs varied and/or converged between the two sites over time. The model outcomes were compared with conceptual and qualitative predictions on benthic community recovery from polymetallic nodule mining published by Jumars (1981). Additionally, it was investigated how $\Delta T..$ developed over time to infer the recovery rate of C flows from experimental deep-sea disturbance in the Peru Basin.




## 2 Methods

### 2.1 Data availability

Macrofauna, megafauna and fish density data (mean±std; ind. m$^{-2}$) for the first four cruises (PD$_{0.1}$ to PD$_7$) were extracted from the original papers (Bluhm, 2001 annex 2.8; Borowski, 2001; Borowski and Thiel, 1998) and methodological details can be

found in those papers. In brief, macrofauna samples (>500 µm size fraction) were collected with a 0.25 m$^{-2}$ box-corer and densities of megafauna and fish were assessed on still photos and videos taken with a towed "Ocean Floor Observation System" (OFOS) underwater camera system. During the PD$_{26}$ cruise (RV Sonne cruise SO242-2; Boetius, 2015), macrofauna were collected with a square $50 \times 50 \times 60$ cm box-corer (disturbed sites: n = 3; undisturbed sites: n = 7) and the upper 5 cm of sediment was sieved on a 500 µm sieve (Greinert, 2015). All organisms retained on the sieve were preserved in 96% un-

denatured ethanol on board (Greinert, 2015) and were sorted and identified ashore to the same taxonomic level as the previous cruises under a stereomicroscope. Megafauna and fish density during the PD$_{26}$ cruise was acquired by deploying the OFOS (Boetius, 2015). Every 20 s, the OFOS automatically took a picture of the seafloor at an approximate altitude of 1.5 m above the seafloor (Boetius, 2015; Stratmann et al., in review) resulting in 1,740 images of plough marks (disturbed sites) and 6,624 images from undisturbed sites (Boetius, 2015). A subset of 300 pictures from the disturbed sites (surface area:

1,440.6 m$^2$) and 300 pictures from the undisturbed sites (surface area: 1,420.4 m$^2$) were randomly selected from the original set of pictures and annotated using the open-source annotation software PAPARA(ZZ)I (Marcon and Purser, 2017). Megafauna were identified to the same taxonomic levels as for the previous megafauna studies conducted within the DEA (Bluhm, 2001), whereas fish were identified to genus level using the CCZ-species atlas (www.ccfzatlas.com).

The above-mentioned density data collected for macrofauna, megafauna and fish were used to build food web models to resolve carbon fluxes; hence, all faunal density data needed conversion into carbon units before they can be used in the food web model. Converting density data to carbon biomass values was challenging in the current study, as few to no conversion factors for deep-sea fauna are available in the literature. Below, we describe the approach we used to tackle this hurdle for macrofauna, megafauna and fish.

In case of a macrofaunal specimen, measuring the carbon content requires its complete combustion, which means that the specimen cannot be kept as voucher specimen in scientific collections. The macrofauna samples collected for this study are part of the Biological Research Collection of Marine Invertebrates (Department of Biology & Centre for Environmental and Marine Studies, University of Aveiro, Portugal) and were therefore not sacrificed. Instead, we used the C conversion factors of macrofauna specimens previously collected within the framework of a pulse-chase experiment in the Clarion-Clipperton

Zone (CCZ, NE Pacific), in which a deep-sea benthic lander (3 incubation chambers à $20 \times 20 \times 20$ cm) was deployed at water depths between 4050 and 4200 m (Sweetman et al., in review). The upper 5 cm of the sediment of the incubation chambers was sieved on 300 µm sieve and preserved in 4% buffered formaldehyde solution. Ashore, the samples were sorted and identified under a dissecting microscope and the biomass of individual freeze-dried, acidified specimens was determined with

at Thermo Flash EA 1112 elemental analyser (EA; Thermo Fisher Scientific, USA) to give the individual carbon content in mmol C ind$^{-1}$. The macrofauna density data (ind. m$^{-2}$) from all cruises were converted to macrofauna biomass (mmol C m$^{-2}$) by multiplying each taxon-specific density (ind. m$^{-2}$) with the mean taxon-specific individual biomass value for macrofauna (mmol C ind$^{-1}$; Table 1). Subsequently, the biomass data of all taxa with the same feeding type (Table 1) were summed to

calculate the biomass of each macrofaunal compartment (mmol C m$^{-2}$; Supplement 1, Figure 2).

The megafauna density data (ind. m$^{-2}$) of the time series was converted to biomass (mmol C m$^{-2}$) by multiplying the taxon-specific density with a taxon-specific mean biomass per megafauna specimen (mmol C ind$^{-1}$; Table 1). To determine this taxon-specific biomass per megafauna specimen, size measurements were used as follows. The 'AUV Abyss' (Geomar Kiel)

equipped with a Canon EOS 6D camera system with 8-15 mm f4 fisheye zoom lens and 24 LED arrays for lightning (Kwasnitschka et al., 2016) flew approximately 4.5 m above the seafloor at a speed of 1.5 m s$^{-1}$ and took one picture every second (Greinert, 2015). Machine vision processing was used to generate a photo-mosaic (Kwasnitschka et al., 2016). A subsample covering an area of 16,206 m$^2$ of the mosaic was annotated using the web-based annotation software 'BIIGLE 2.0' (Langenkämper et al., 2017). The length of all megafauna taxa for which data were available from previous cruises was

measured using the approach presented in Durden et al. (2016). Briefly, depending on the taxon, either body length, the diameter of the disk, or the length of an arm were measured on the photo-mosaic and converted into biomass per individual (g ind$^{-1}$) using the relationship between measured body dimensions (mm) and preserved wet weight (g ind$^{-1}$) (Durden et al., 2016). Subsequently, the preserved wet weight (g ind$^{-1}$) was converted to fresh wet weight (g ind$^{-1}$) using conversion factors from Durden et al. (2016) and to organic carbon (g C ind$^{-1}$ and mmol C ind$^{-1}$) using the taxon-specific conversion factors

presented in Rowe (1983). For the taxa Cnidaria and Porifera no conversion factors were available. Therefore, taxon-specific individual biomass values were extracted from a study from the CCZ (Tilot, 1992). The individual biomass of Bryozoa and Hemichordata were calculated as the average biomass of an individual deep-sea megafauna organism (B, mmol C ind$^{-1}$) at 4100 m depth following from the ratio of the regression for total biomass and abundance by Rex et al. (2006):

$$B = \frac{10^{(-0.734-0.00039 \times \text{depth})}}{10^{(-0.245-0.00037 \times \text{depth})}}. \tag{1}$$

Following the approach applied to the macrofauna dataset, individual biomasses of taxa with similar feeding types (Table 1) were summed to determine the biomass of the megafauna food-web compartments (mmol C m$^{-2}$; Supplement 1; Figure 1).

Individual biomass of fish was calculated using the allometric relationship for *Ipnops agassizii*:

wet weight = a $\times$ length$^b$, $\tag{2}$

where a = 0.0049 and b = 3.03 (Froese and Pauly, 2017; Froese et al., 2014), as *Ipnops* sp. was the most abundant deep-sea fish observed at the DEA (60% of total fish density at undisturbed and 40% of total fish density at disturbed sites). The length (mm) of all *Ipnops* sp. specimens was measured on the annotated 600 pictures (300 pictures from undisturbed site, 300 pictures from disturbed site) in PAPARA(ZZ)I (Marcon and Purser, 2017) using three laser points captured in each image (distance





between laser points: 0.5 m (Boetius, 2015)). The wet weight (g) was converted to dry-weight and subsequently to carbon content (mmol C ind$^{-1}$) using the taxon-specific conversion factors presented in Brey et al. (2010).

## 2.2 Food web structure

The faunal biomass was further divided into feeding guilds in order to define the food web compartments of the model. Fish (Osteichthyes) were classified as scavenger/ predator and invertebrate macrofauna and megafauna were divided into filter/suspension feeders (FSF), deposit feeders (DF), carnivores (C) and omnivores (OF) (Figure 2). Since feeding types are well described for polychaetes (Jumars et al., 2015), we made a further detailed classification of the macrofaunal polychaetes into suspension feeders (PolSF), surface deposit feeders (PolSDF), subsurface deposit feeders (PolSSDF), carnivores (PolC), and omnivores (PolOF).

External carbon sources that were considered in the model included suspended detritus in the water column (Det_w), labile (lDet_s) and semi-labile detritus (sDet_s) in the sediment. Suspended detritus was considered a food source for polychaete, macrofaunal and megafaunal suspension feeders. Labile and semi-labile sedimentary detritus was a source for deposit-feeding and omnivorous polychaetes, macrofauna and megafauna. Omnivores and carnivores of each size class preyed upon organisms of the same and smaller size classes, i.e. MegC and MegOF preyed upon MegDF, MegFSF, MacFSF, MacDF, MacC, MacOF, PolSDF, PolSSDF, PolSF, PolOF, and PolC. Furthermore, MacC, PolC, MacOF, and PolOF preyed upon MacFSF, MacDF, PolSDF, PolSSDF, and PolSF. Fish preyed upon all fauna and the carcass pool. This carcass pool consisted of all fauna (macrofauna, megafauna and fish) that died in the food web and was also the food source of omnivores.

Carbon losses from the food web were respiration to dissolved inorganic carbon (DIC), predation on macrofauna, megafauna and fish by pelagic/ benthopelagic fish, scavenging on carcasses by pelagic/ benthopelagic scavengers and faeces production by all faunal compartments.

## 2.3 Literature constraints

The carbon flows between faunal compartments are constrained by the implementation of various minimum and maximum process rates and conversion efficiencies as inequalities in all models, which are described here. Assimilation efficiency (AE) is calculated as:

$$AE = (I-F) / I, \tag{3}$$

where I is the ingested food and F are the faeces (Crisp, 1971). The min-max range was set from 0.62 to 0.87 for macrofauna and polychaetes (Stratmann et al., in prep.), from 0.48 to 0.80 for megafauna (Stratmann et al., in prep.) and from 0.84 to 0.87 for fish (Drazen et al., 2007).

Net growth efficiency (NGE) is defined as:

$$NGE = P / (P + R), \tag{4}$$





with P being secondary production and R being respiration (Clausen and Riisgård, 1996). The min-max ranges are set to 0.60 to 0.72 for macrofauna and polychaetes (Clausen and Riisgård, 1996; Navarro et al., 1994; Nielsen et al., 1995), from 0.48 to 0.60 for megafauna (Koopmans et al., 2010; Mondal, 2006; Nielsen et al., 1995) and from 0.37 to 0.71 for fish (Childress et al., 1980). The secondary production P (mmol C m$^{-2}$) is calculated as:

P = P/B-ratio × biomass,                                                                                      (5)

with the P/B-ratios for macrofauna and polychaetes ($8.49 \times 10^{-4}$ to $4.77 \times 10^{-3}$ d$^{-1}$; (Stratmann et al., in prep.)), megafauna ($2.74 \times 10^{-4}$ to $1.42 \times 10^{-2}$ d$^{-1}$; (Stratmann et al., in prep.)) and fish ($6.30 \times 10^{-4}$ d$^{-1}$; (Collins et al., 2005; Randall, 2002)). The respiration rate R (mmol C m$^{-2}$) was calculated as:

R = bsFR × biomass,                                                                                              (6)

where bsFR is the biomass-specific fauna respiration rate (d$^{-1}$) and ranges were fixed between $7.12 \times 10^{-5}$ to $2.28 \times 10^{-2}$ d$^{-1}$ for macrofauna and polychaetes (Stratmann et al., in prep.), $2.74 \times 10^{-4}$ to $1.42 \times 10^{-2}$ d$^{-1}$ for megafauna (Stratmann et al., in prep.) and $2.3 \times 10^{-4}$ and $3.6 \times 10^{-4}$ d for fish (Mahaut et al., 1995; Smith and Hessler, 1974).

## 2.4 Linear inverse model solution and network index

A food web model with all compartments present in the food web, like e.g. the PD$_{26}$ food web model for the undisturbed site,
consists of 147 carbon flows with 14 mass balances, i.e. food-web compartments, and 76 data inequalities leading to a mathematically under-determined model (14 equalities vs. 147 unknown flows). Therefore, the LIMs were solved with the R package 'LIM' (van Oevelen et al., 2010) in R (R-Core-Team, 2016) following the likelihood approach (van Oevelen et al., 2010) to quantify the mean and standard deviations of each of the carbon flows from a set of 100,000 solutions. This set was sufficient to guarantee the convergence of mean and standard deviation within a 2.5% deviation.

The network index 'total system throughput' ($T..$) was calculated with the R-package 'NetIndices' (Kones et al., 2009) for each of the 100,000 model solutions and subsequently summarized as mean ± standard deviation.

## 2.5 Statistical analysis

Statistical differences between compartment biomasses of the undisturbed vs. disturbed sites for the same sampling event (PD$_{0.1}$, PD$_{0.5}$, PD$_3$, and PD$_7$; PD$_{26}$ was omitted due to a lack of megafauna replicates) were assessed by calculating Hedge's d
(Hedges and Olkin, 1985a), which is especially suitable for small sample sizes (Koricheva et al., 2013):

$d=(\bar{Y}^E-\bar{Y}^C)/(((n^E-1)(s^E)^2+(n^C-1)(s^C)^2)/(n^E+n^C-2))^{0.5} \times J$                             (7)

with $J=1-(3/(4(n^E + n^C-2)-1))$,                                                                              (8)

where $\bar{Y}^E$ is the mean of the experimental group (i.e. the biomass at disturbed sites of a particular year), $\bar{Y}^C$ is the mean of the control group (i.e. the biomass at undisturbed sites of the respective year), $s^E$ and $s^C$ are the standard deviations with
corresponding groups, $n^E$ and $n^C$ are the sample sizes of the corresponding groups. The variance of Hedge's d $\sigma_d^2$ (Koricheva et al., 2013) is estimated as:

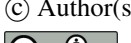

$$\sigma_d{}^2 = (n^E + n^C)/(n^E n^C) + d^2/(2(n^E + n^C)). \tag{9}$$

The weighted Hedge's d and the estimated variance (Hedges and Olkin, 1985b) of the total biomass of all compartments of the same sampling event were calculated as:

$$d{+} = sum(d_i/\sigma_{di}{}^2)/sum(1/\sigma_{di}{}^2), \tag{10}$$

with $\sigma_{d+}{}^2 = 1/sum(1/\sigma_{di}{}^2)$.

Following Cohen (1988)'s rule of thumb for effect sizes, Hedge's d=|0.2| signifies a small experimental effect, implying that the biomass of the food-web compartments is similar between the disturbed and undisturbed sites. When Hedge's d=|0.5|, the effect size is medium, hence there is moderate difference, and when Hedge's d=|0.8|, the effect size is large, i.e. there is a large difference between the biomass of the compartments between sites.

The network index $T..$ was compared between the undisturbed and disturbed sites of the same sampling event by assessing the fraction of the $T..$ values of the 100,000 model solutions of the undisturbed food web that were larger than the $T..$ values of the 100,000 model solutions of the disturbed food web. When this fraction is >0.95, the difference in 'total system throughput' between the two food-webs from the same sampling event is considered significantly different (van Oevelen et al., 2011), indicating that the carbon flows in the food web from that specific sampling event have not recovered from the experimental disturbance.

## 3 Results

### 3.1 Food-web structure and trophic composition

Total faunal biomass was always higher at the undisturbed sites as compared to the disturbed sites from the same sampling year (Figure 1, Supplement 1), and ranged from a minimum of 5.45±1.27 mmol C m$^{-2}$ (PD$_{0.1}$) to a maximum 22.33±3.40 mmol C m$^{-2}$ (PD$_3$) at the undisturbed sites and from minimum of 1.36±1.24 mmol C m$^{-2}$ (PD$_{0.1}$) to maximum 15.82±1.99 mmol C m$^{-2}$ (PD$_3$) at the disturbed sites. At PD$_{0.1}$ the total faunal biomass at the disturbed sites was only 25% of the total faunal biomass at the undisturbed sites, whereas at PD$_3$ the total faunal biomass at the disturbed sites was 71% of the total faunal biomass at the undisturbed sites. At PD$_{26}$, the faunal biomass at the disturbed sites was 54% of the biomass at the undisturbed sites. The absolute weighted Hedge's d |d$_+$| of all faunal compartment biomasses for PD$_{0.1}$ to PD$_7$ ranged from 0.053±0.019 at PD$_{0.5}$ to 0.075±0.019 (Supplement 2), indicating a strong experimental effect and therefore that biomasses of all faunal compartment did not recover over the period analysed (PD$_{0.1}$ to PD$_7$).

The faunal biomass at both the undisturbed and disturbed sites from PD$_{0.1}$ to PD$_7$ was dominated by deposit feeders (from 63% at undisturbed PD$_{0.1}$ to 83% at disturbed PD$_{0.5}$ and disturbed PD3) (Figure 3). In contrast, at the undisturbed sites of PD$_{26}$, the largest contribution to total faunal biomass was from filter- and suspension feeders (44%), whereas deposit feeders only contributed 35%. At the disturbed sites of PD$_{26}$, deposit feeders had the highest biomass (61%), followed by carnivores (19%) and filter- and suspension feeders (14%).



## 3.2 Carbon flows

The total faunal C ingestion (mmol C $m^{-2}$ $d^{-1}$) ranged from $8.63\times10^{-3}\pm1.58\times10^{-5}$ at the disturbed sites at $PD_{0.1}$ to $1.47\times10^{-1}\pm8.55\times10^{-4}$ at the undisturbed sites at $PD_3$ and was always lower at the disturbed sites compared to the undisturbed sites (Figure 4A; Supplement 3). The ingestion consisted mainly of the sedimentary detritus (labile and semi-labile) that contributed

between 56.97% (undisturbed sites, $PD_{26}$) and 99.50% (disturbed sites, $PD_{0.1}$) to the total carbon ingestion.

Faunal respiration (mmol C $m^{-2}$ $d^{-1}$) ranged from $6.02\times10^{-3}\pm6.75\times10^{-5}$ (disturbed sites, $PD_{0.5}$) to $3.92\times10^{-2}\pm3.69\times10^{-4}$ (undisturbed sites, $PD_3$). During the twenty-six years after the DISCOL experiment, modelled faunal respiration was always higher at undisturbed sites as compared to disturbed sites (Table 2, Figure 4). Over time, non-polychaete macrofauna contributed least to total faunal respiration (Table 2), except at the disturbed sites of $PD_{0.5}$ and at both sites of $PD_3$. During this

$PD_3$ sampling campaign, macrofauna contributed 49.97% at the undisturbed sites and 58.35% at the disturbed sites to the total faunal respiration. Polychaetes respired between 18.59% of the total fauna respiration at the undisturbed sites at $PD_{26}$ and 77.61% of the total fauna respiration at the disturbed sites at $PD_{0.5}$. The megafauna respiration contribution was highest at $PD_{26}$, where they respired 64.95% of the total faunal respiration at the disturbed sites and 78.67% of the total faunal respiration at the undisturbed sites. The contribution of fish to total faunal respiration was always <2%. Besides respiration, faeces

production contributed between 20.07% at disturbed $PD_3$ and 34.65% at disturbed $PD_{0.1}$ to total carbon outflow from the food web (Figure 4). The contribution of the combined outflow of predation by external predators and scavengers on carcasses to the total C loss from the food web ranged from 50.48% at disturbed $PD_7$ to 65.33% at disturbed $PD_{0.1}$.

The fraction of $T_{..}$ values that were larger for the food webs at the undisturbed sites than for the disturbed sites from the same sampling event was 1.0 at $PD_{0.1}$, $PD_{0.5}$, $PD_3$, $PD_7$ and $PD_{26}$. No decreasing trend in $\Delta T_{..}$ over time was visible (Figure 5), in

fact, the largest $\Delta T_{..}$ were calculated for $PD_3$ ($7.87\times10^{-2}\pm1.97\times10^{-3}$ mmol C $m^{-2}$ $d^{-1}$) and $PD_{26}$ ($7.67\times10^{-2}\pm9.41\times10^{-4}$ mmol C $m^{-2}$ $d^{-1}$).

## 4 Discussion

This study assessed the evolution of the food web structure and ecosystem function 'faunal C cycling' in an abyssal nodule-rich soft-sediment ecosystem following an experimental sediment disturbance. By comparing a time-series over 26 years with

food web models (undisturbed vs. disturbed sites), we show that the total faunal biomass at the disturbed site was still only about half of the total faunal biomass at the undisturbed sites 26 years after the disturbance. Furthermore, the role of the various feeding types in the carbon cycling differs and the 'total system throughput' $T_{..}$, i.e. the sum of all carbon flows in the food web, was still significantly lower at the disturbed sediment compared to the undisturbed sediment after 26 years.



## 4.1 Model limitations

Our results are unique as it allowed for the first time to assess the recovery of C cycling in benthic deep-sea food webs from a small-scale sediment disturbance in polymetallic nodule rich areas. However, the models come with limitations. The standard procedures to assess megafauna densities have evolved during the 26 years of post-disturbance monitoring. The OFOS system

used 26 years after the initial DISCOL experiment took pictures automatically every 20 s from a distance of 1.5 m above the seafloor (Boetius, 2015; Stratmann et al., in review). By contrast, the OFOS system used in former cruises was towed approximately 3 m above the seafloor and pictures were taken selectively by the operating scientists (Bluhm and Gebruk, 1999). Therefore, the procedure used in the former cruises very likely led to an overestimation of rare and charismatic megafauna, and probably to an underestimation of dominant fauna and organisms of small size (<3 cm) for $PD_{0.1}$ to $PD_7$ as

compared to $PD_{26}$.

Previous cruises to the DEA focused on monitoring changes in faunal density and diversity, but not on changes in biomass. Hence, a major task in this study was to find appropriate conversion factors to convert density into biomass. However, no individual biomass data for macrofauna taxa were available for the Peru Basin, so we used data from sampling stations of similar water depths in the eastern Clarion-Clipperton Zone (CCZ, NE Pacific; Sweetman et al., in review). As organisms in

deep-sea regions with higher organic carbon input are larger than their counterparts from areas with lower organic carbon input (McClain et al., 2012), using individual biomass data from the CCZ, a more oligotrophic region than the Peru Basin (Haeckel et al., 2001; Vanreusel et al., 2016) might have led to an underestimation of the biomass for macrofauna. However, this has likely limited impact on the interpretation of the comparative results within the time series, because the same methodology was applied throughout the time series dataset. Moreover, the determination of megafauna biomass was also difficult as no

size measurements were taken from megafauna individuals during the $PD_{0.1}$ to $P_{D7}$ cruises. Consequently, it was not possible to detect differences in size classes between disturbed and undisturbed sediments or recruitment events in e.g. echinoderms (Ruhl, 2007) following the DISCOL experiment. Instead, we used fixed conversion factors for the different taxa for the entire time series.

## 4.2 Feeding-type specific differences in recovery

Eight years before the experimental disturbance experiment was conducted at the DISCOL area, Jumars (1981) qualitatively predicted the response of different feeding types in the benthic community to polymetallic nodule removal. Although several seabed test mining or mining simulations were performed since then (Jones et al., 2017), no study compared or verified these conceptual predictions on feeding-type specific differences in recovery from deep-sea mining. As few comparative studies are available, we compare here our food-web model results with those of the conceptual model predictions for scavengers, surface

and subsurface deposit feeders and suspension feeders by Jumars (1981).



Jumars (1981) predicted that organisms inside the mining tracks would be killed either by the fluid shear of the dredge/ plough or by abrasion and increased temperatures inside the rising pipe with a mortality rate of >95%. In contrast, the impact on mobile and sessile organisms in the vicinity of the tracks would depend on their feeding type (Jumars, 1981).

The author also predicted that the density of mobile scavengers, such as fish and lysianassid amphipods would rise shortly

after the disturbance in response to the increased abundance of dying or dead organisms within the mining tracks. Indeed, when plotting the respiration of fish (in mmol C $m^{-1}$ $d^{-1}$) normalized to the fish respiration at the undisturbed sediment at $PD_{0.1}$ over time, the respiration for the undisturbed sediment increased steeply until $PD_3$ and dropped subsequently (Figure 6). However, experiments with baits at PAP and the Porcupine Seabight (NE Atlantic) showed that the scavenging deep-sea fish *Coryphaenoides armatus* intercept bait within 30 min (Collins et al., 1999) and stayed at the food fall for 114±55 min (Collins

et al., 1998). Hence, it is very likely that this rise in fish respiration at the undisturbed sediment 0.5 years after the DISCOL is a result of natural variability as opposed to the predicted rise in scavenger density and/ or biomass caused by the mining activity. At the disturbed sediment, no fish were detected at $PD_{0.1}$ or $PD_{0.5}$, which could be related to lack of prey in a potential predator-prey relationship (Bailey et al., 2006). However, because of the relatively small area of disturbed sediment (only 22% of the 10.8 $km^2$ of sediment were ploughed (Thiel and Schriever, 1989)), the low density of deep-sea fish (e.g. between 7.5

and 32 ind. $ha^{-1}$ of the dominant fish genus *Coryphaenoides* sp. at Station M (Bailey et al., 2006)) and the high motility of fish, this observation may be coincidental.

Jumars (1981) predicted that, on a short term, subsurface deposit feeders outside the mining tracks would be the least impacted feeding type, because of their relative isolation from the re-settled sediment, and their relative independence of organic matter on the sediment surface, whereas subsurface deposit feeders inside the mining tracks would experience high mortality. For the

long-term recovery, the author pointed to the dependence of subsurface deposit feeders on bacterial production in the sediment covered with re-resettled sediment. In our food web model, sub-surface and surface deposit feeders were grouped into the deposit feeder category, except for polychaetes, for which we kept the surface-subsurface distinction. The biomass of PolSSDF fluctuated by one order of magnitude over the 26-year time series and had high biomass values at the undisturbed $PD_{0.1}$ site, the disturbed $PD_3$ sites and at both sites at $PD_7$. The normalized respiration of PolSSDF also showed strong fluctuations at the

undisturbed and disturbed sites over time (Figure 6) indicating a large natural variability or variable sampling results. Such temporal dynamics in deep-sea macrofauna were detected at Station M, where the density of several dominating metazoan macrofauna increased eight months after a peak in POC flux was measured at 50 and 600 m above the seafloor (Drazen et al., 1998). Hence, Jumars (1981) predictions for sub-surface deposit feeders could not be tested, provided the natural fluctuations in PolSSDF densities that were used to calculate biomass.

Jumars (1981) anticipated that surface deposit feeders would suffer more strongly from deep-sea mining activities compared to sub-surface deposit feeders because the rate of sediment deposition would increase inside and beyond mining tracks, with this newly settling sediment altering the sediment composition and food concentration in the sediment. Indeed, the recovery of holothurian densities at the DEA was probably delayed owing to unfavourable food conditions (Stratmann et al., in review). Nevertheless, deposit feeders seem to have advantages during the recovery from the DISCOL disturbance experiment. When




comparing the contribution of deposit feeders from all size classes (macrofauna, polychaetes, megafauna) to respiration, predation by external predators and faeces production to the contribution of omnivores, filter- and suspension feeders and carnivores, their contribution was always higher at the disturbed site compared to the undisturbed site of the same sampling event. However, owing to the overall lower biomass inside the disturbed area compared to the undisturbed area, the absolute

carbon respiration (in mmol C m$^{-2}$ d$^{-1}$) remained lower for deposit feeders at the disturbed site compared to the corresponding undisturbed site, even after 26 years when this difference was 2.6%.

Jumars (1981) expected that the suspension feeders outside the mining tracks would be negatively affected during the presence of the sediment plumes and/ or as long as their filtration apparatus was clogged by sediment. This "clogging" hypothesis could not be tested here, because the models did not resolve these unknown changes in faunal physiology, but could only assess

carbon cycling differences associated with differences in biomass. Furthermore, Jumars (1981) anticipated that the recovery of nodule-associated organisms, such as filter and suspension feeding Porifera, Antipatharia or Ascidiacea (Vanreusel et al., 2016) would require more than 10,000 years, owing to the slow growth rate of polymetallic nodules (Guichard et al., 1978; Kuhn et al., 2017) and the removal and/ or burial of the nodules. Directly after the initial DISCOL disturbance event, the respiration rate of filter and suspension feeders at the disturbed sediment was only 1% of the respiration rate of this feeding

type at the undisturbed sediment. After 26 years, the relative difference in the filter and suspension feeding respiration rate was still 80%. Part of this difference at PD$_{26}$ resulted from the presence of a single specimen of Alcyonacea with a biomass of 4.71 mmol C m$^{-2}$ at the undisturbed site. However, even if we ignore this Alcyonacea specimen in the model, the respiration of suspension and filter feeding in the disturbed site would still be 71% lower compared to the undisturbed site, indicating a slow recovery of this feeding group.

To summarize the comparison of modelled potential recovery of the different feeding types with the predictions by Jumars (1981), scavenging and predatory fish at the undisturbed sediment followed first the predicted density pattern, though this might also have been related to natural variability. After three years, however, the fish contribution to carbon cycling was lower than expected from the predictions. Owing to an apparently strong natural variability in polychaete subsurface deposit feeder biomass, the recovery prognosis for subsurface deposit feeders could not be tested. Furthermore, it could not be assessed

whether surface deposit feeders were more strongly affected by the mining activity than subsurface deposit feeders. In general, the time series analysis showed that deposit feeders likely benefited from the disturbance experiment in comparison to other feeding types. Confirming Jumars (1981) prediction, the activity of filter and suspension feeders in the food web did not recover within 26 years.

## 5 Conclusion

Deep-sea mining will negatively impact the benthic ecosystem of abyssal ecosystems. It is therefore important to be able to estimate how long the recovery of the ecosystem after a deep-sea mining operation will take. This study used the linear inverse modelling technique to compare the carbon flows between different food web compartments at undisturbed and disturbed sites



at the DISCOL experimental area in the Peru Basin over a period of 26 years. Even after 26 years, the total faunal biomass and the total food-web activity (i.e. summed carbon cycling) at the disturbed sites was only approximately half (54% and 56% respectively) of the total faunal biomass and food-web activity at the undisturbed sites. Deposit feeders were the least impacted by the sediment disturbance, with less than 3% relative difference in total carbon loss (i.e. respiration, external predation and

feces production) between undisturbed and disturbed sites after 26 years. In contrast, filter and suspension feeders did not recover at all and the relative difference in respiration rate was 79%. Overall, it can be concluded that ecosystem functioning (as measured by total carbon cycling) within the macrofauna, megafauna and fish has not recovered 26 years after the experimental disturbance.

**Data availability**

Data on biomass of the different food web compartments are presented in Supplement 1. Data on Hedge's d, the corresponding standard deviations, weighted Hedge's d and weighted standard deviation are presented in Supplement 2. The mean and standard deviations calculated for each carbon flux over 100,000 iterations for all food webs from the undisturbed and disturbed site for all time steps is presented in Supplement 3. All OFOS images associated with this article are available from the PANGAEA storage archive.

**Authors contribution**

TS went through the published literature for data input to the model, LL, AP, YM, CR, AR, MRC, ESL, AKS, DOBJ and KK contributed data, TS and DvO developed the food web models, TS and DvO wrote the manuscript with input from all co-authors.

**Acknowledgements**

We thank the chief scientists Jens Greinert (SO242-1) and Antje Boetius (SO242-2) as well as captain and crew of RV Sonne for their excellent support during both legs of cruise SO242. We also thank the 'AUV Abyss' team from Geomar, Kiel (Germany) and Daniëlle de Jonge (Groningen University, The Netherlands) for identifying the fish species. The research leading to these results has received funding from the European Union Seventh Framework Programme (FP7/2007-2013) under the MIDAS project, grant agreement n° 603418 and by the JPI Oceans – Ecological Aspects of Deep Sea Mining project

(NWO-ALW grant 856.14.002) and the Bundesministerium für Bildung und Forschung (BMBF) grant n° 03F0707A-G. Further financial support was granted to CESAM (UID/AMB/50017 - POCI-01-0145-FEDER-007638), to FCT/MCTES by national funds (PIDDAC), and by co-funding by the FEDER, within the PT2020 Partnership Agreement and Compete 2020. CFR was supported by Fundação para a Ciência e a Tecnologia (FCT) grant (SFRH/ BPD/107805/2015).





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




**Table 1.** Taxon-specific biomass per individual (mmol C ind$^{-1}$) for macrofauna and megafauna including the specific feeding types. Macrofauna biomass data are based on macrofauna specimen collected in the abyssal plains of the Clarion-Clipperton Zone (NE Pacific) (Sweetman et al., in review). In contrast, megafauna biomass was estimated by converting size-measurements of specific body parts of organisms from DEA that were acquired using photo-annotation into preserved wet

5   weight per organism using the relationships presented in Durden et al. (2016). Subsequently the preserved wet weight was converted into fresh wet weight and biomass following the conversions presented in Durden et al. (2016) and Rowe (1983). Whenever no conversion factors for a specific taxon were reported in Durden et al. (2016) mean taxon-specific biomass data per individual were extracted from Tilot (1992) for the CCZ.

The abbreviation are: C = carnivores, DF = deposit feeders, FSF = filter/ suspension feeders, O = omnivores, PolC =

10  carnivorous polychaete, PolOF = omnivorous polychaete, PolSF = suspension feeding polychaete, PolSDF = surface deposit feeding polychaete, PolSSDF = subsurface deposit feeding polychaete, S = scavengers.

References: [1](Fox et al., 2003), [2](Menzies, 1962), [3](McClain et al., 2012), [4](Smith and Stockley, 2005), [5](Gage and Tyler, 1991), [7](Jumars et al., 2015), [8](Bluhm, 2001), [9](Drazen and Sutton, 2017)

| Size class | Taxon | Feeding type | n | Biomass (mmol C ind$^{-1}$) (Mean±Std) |
|---|---|---|---|---|
| Macro-fauna | Bivalvia[a] | FSF[1] | 7 | $1.41\times10^{-3}\pm8.29\times10^{-4}$ |
| | Cumacea[a] | DF[1] | 2 | $3.09\times10^{-3}\pm6.22\times10^{-4}$ |
| | Echinoidea[b] | 85% O, 15% DF[4] | 64 | $9.66\times10^{-3}\pm2.84\times10^{-2}$ |
| | Gastropoda[a] | 90% DF, 10% C[3] | 2 | $8.56\times10^{-2}\pm3.98\times10^{-2}$ |
| | Isopoda[a] | 93% DF, 7% C[2] | 4 | $1.33\times10^{-3}\pm1.06\times10^{-3}$ |
| | Ophiuroidea[b] | C[1] | 64 | $9.66\times10^{-3}\pm2.84\times10^{-2}$ |
| | Polychaeta[a] | PolSF, PolSDF, PolSSDF, PolC, PolOF[7] | 26 | $1.33\times10^{-2}\pm3.68\times10^{-2}$ |
| | Scaphopoda[b] | C[1] | 64 | $9.66\times10^{-3}\pm2.84\times10^{-2}$ |
| | Tanaidacea[a] | DF[1] | 5 | $5.48\times10^{-3}\pm1.04\times10^{-2}$ |
| Mega-fauna | Actiniaria | FSF[1] | 301 | $2.95\times10^{-1}\pm8.75\times10^{-1}$ |
| | Antipatharia | FSF[1] | 3 | 177.30±68.23 |
| | Ascidiacea[d] | FSF[1] | | $8.30\times10^{-1}$ |
| | Asteroidea | C[1] | 53 | 139.23±43.56 |
| | Bryozoa[g] | FSF[1] | | 22.38 |
| | Cephalopoda | C[1] | 7 | 46.85±27.88 |





| | Taxon | | | |
|---|---|---|---|---|
| | Ceriantharia[d] | FSF[1] | | 1923.17 |
| | Cnidaria[c] | FSF[1] | | $2.35 \times 10^{-1}$ |
| | Crinoidea[d] | FSF[1] | | 5.33 |
| | Crustacea | C[1,8] | 541 | 2.56±10.05 |
| | Echinoidea[d] | 15% DF, 85% OF[4] | | 59.17 |
| | Alcyonacea[d] | FSF[1] | | 21.67 |
| | Hemichordata[g] | DF[5,8] | | 22.38 |
| | Holothuroidea[e] | DF[1] | 450 | 154.32±332.51 |
| | Ophiuroidea | C[1] | 527 | 16.05±10.15 |
| | Pennatularia[d] | FSF[1] | | 21.67 |
| | Polychaeta | PolSF, PolSDF, PolSSDF, PolC, PolOF[7] | 62 | $5.30 \times 10^{-1} \pm 1.20 \times 10^{-2}$ |
| | Porifera[c] | FSF[1] | | 6.74 |
| Fish | Osteichthyes[f] | S, C[9] | 10 | 73.36±41.12 |

[a]Taxon-specific individual biomass; [b]Individual biomass calculated based on all other macrofauna data; [c]Median taxon-specific individual biomass for individuals from the Porcupine Abyssal Plain where Durden et al. (2016) did not have reliable dimension measurements; [d]Mean taxon-specific biomass data per individual were extracted from Tilot (1992) for the CCZ; [e]Individual biomass of *Benthodytes* sp., one of the most abundant holothurian morphotype at the DISCOL site (Stratmann et al., in review); [f]Individual biomass of *Ipnops* sp., the most abundant deep-sea fish at the PD$_{26}$ undisturbed site; [g]Individual biomass calculated for mean benthos megafauna at 4100 m depth based on the biomass-bathymetry and abundance-bathymetry relationships presented in Rex et al. (2006).





**Table 2.** Faunal respiration rate (mmol C m$^{-2}$ d$^{-1}$) and contribution (%) of the size classes macrofauna, polychaetes, megafauna and fish to the respiration for the undisturbed (Undist.) and disturbed (Dist.) sites directly after the disturbance event in March 1989 (PD$_{0.1}$), 0.5 years post-disturbance (September 1989, PD$_{0.5}$), 3 years post-disturbance (January 1992, PD$_3$), 7 years post-disturbance (February 1996, PD$_7$) and 26 years post-disturbance (September 2015, PD$_{26}$).

| | PD$_{0.1}$, Undist. | PD$_{0.1}$, Dist. | PD$_{0.5}$, Undist. | PD$_{0.5}$, Dist. | PD$_3$, Undist. | PD3, Dist. | PD7, Undist. | PD7, Dist. | PD26, Undist. | PD26, Dist. |
|---|---|---|---|---|---|---|---|---|---|---|
| Faunal respiration | $1.02\times10^{-2}\pm$ $1.17\times10^{-4}$ | $2.72\times10^{-3}\pm$ $5.23\times10^{-6}$ | $1.07\times10^{-2}\pm$ $5.73\times10^{-5}$ | $6.02\times10^{-3}\pm$ $6.75\times10^{-5}$ | $3.92\times10^{-2}\pm$ $3.68\times10^{-4}$ | $2.99\times10^{-2}\pm$ $2.33\times10^{-4}$ | $2.14\times10^{-2}\pm$ $2.50\times10^{-4}$ | $1.54\times10^{-2}\pm$ $1.49\times10^{-4}$ | $2.00\times10^{-2}\pm$ $1.50\times10^{-4}$ | $1.13\times10^{-2}\pm$ $1.04\times10^{-4}$ |
| Macrofauna | 8.63 | 7.34 | 9.73 | 14.35 | 49.97 | 58.35 | 6.50 | 4.51 | 2.64 | 1.19 |
| Polychaeta | 61.59 | 77.80 | 62.69 | 77.61 | 27.09 | 30.03 | 67.08 | 83.51 | 18.52 | 32.43 |
| Megafauna | 29.47 | 14.85 | 27.06 | 8.04 | 22.30 | 11.54 | 25.75 | 11.63 | 78.67 | 64.95 |
| Fish | $3.02\times10^{-1}$ | 0.00 | $5.29\times10^{-1}$ | 0.00 | $6.43\times10^{-1}$ | $7.75\times10^{-2}$ | $6.64\times10^{-1}$ | $3.53\times10^{-1}$ | $1.73\times10^{-1}$ | 1.44 |



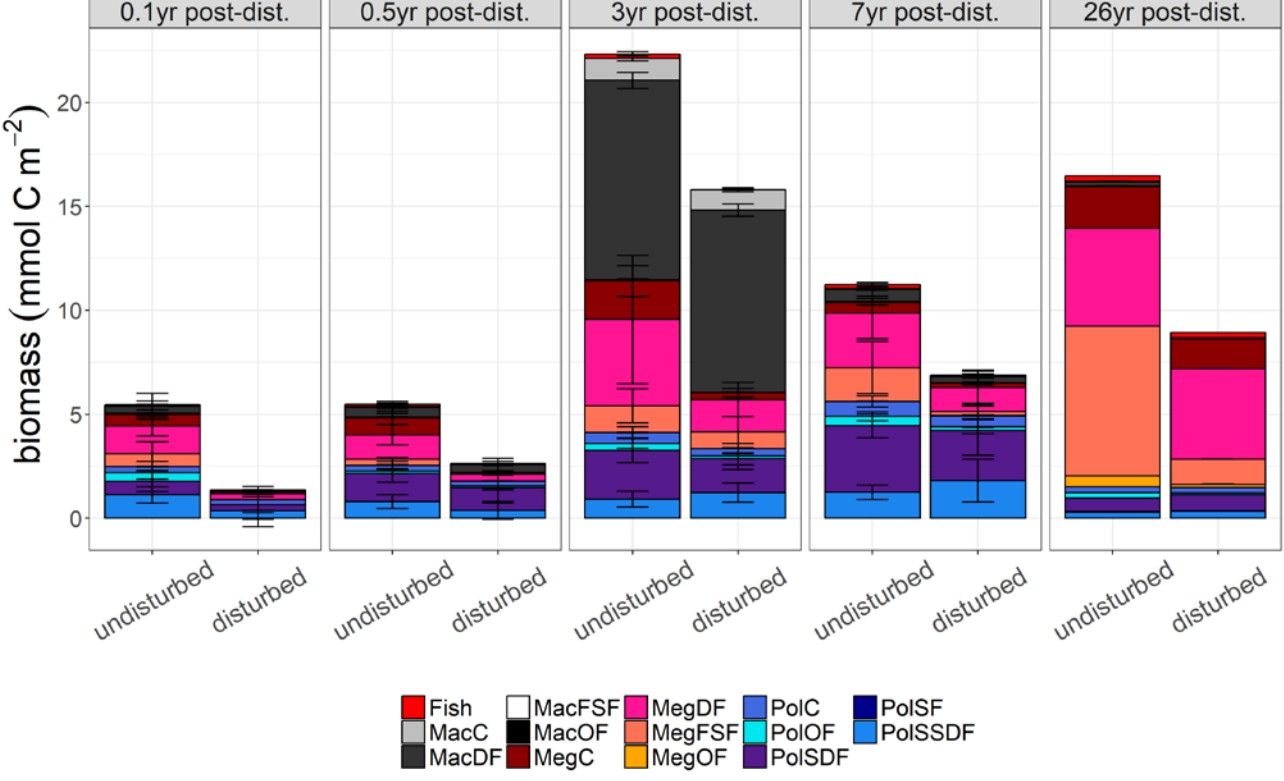

**Figure 1.** Mean biomass (mmol C m$^{-2}$) of the food web compartments for the undisturbed and disturbed sites inside the DISCOL experimental area (Peru Basin, SE Pacific) 0.1 years post-disturbance (PD$_{0.1}$), for 0.5 years post-disturbance (PD$_{0.5}$), for three years post-disturbance (PD$_3$), for seven years post-disturbance (PD$_7$), and for 26 years post-disturbance (PD$_{26}$). The error bars represent 1 standard deviation.

The abbreviation are: MacC = macrofauna carnivores, MacDF = macrofauna deposit feeders, MacFSF = macrofauna filter/ suspension feeders, MacO = macrofauna omnivores, MegC = megafauna carnivores, MegDF = megafauna deposit feeders, MegFSF = megafauna filter/ suspension feeders, MegOF = megafauna omnivores, PolC = polychaete carnivores, PolOF = polychaete omnivores, PolSDF = polychaete surface deposit feeders, PolSF = polychaete suspension feeders, PolSSDF = polychaete subsurface deposit feeders.

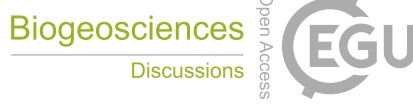

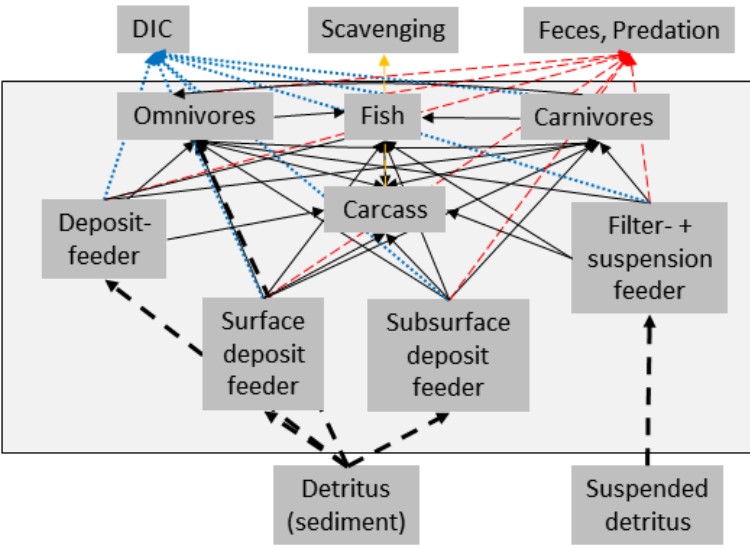

**Figure 2.** Simplified schematic representation of the food web structure that forms the basis of the linear inverse model (LIM). All compartments inside the box were part of the food web model, whereas compartments outside the black box were only considered as carbon influx or efflux, but were not directly modelled. In order to simplify the graph, for macrofauna, polychaetes and megafauna, only feeding types were presented and no size classes. Solid black arrows represent the carbon flux between food-web compartments and black dashed arrows represent the influx of carbon to the model. Blue-dotted arrows show the loss of carbon from the food web via respiration to DIC. The red dashed arrows indicate the loss of carbon from the food web as faeces and as predation by pelagic/ benthopelagic fish and the yellow-dashed arrow indicate the reduction of the carcass pool due to scavenging by pelagic/ benthopelagic fish.



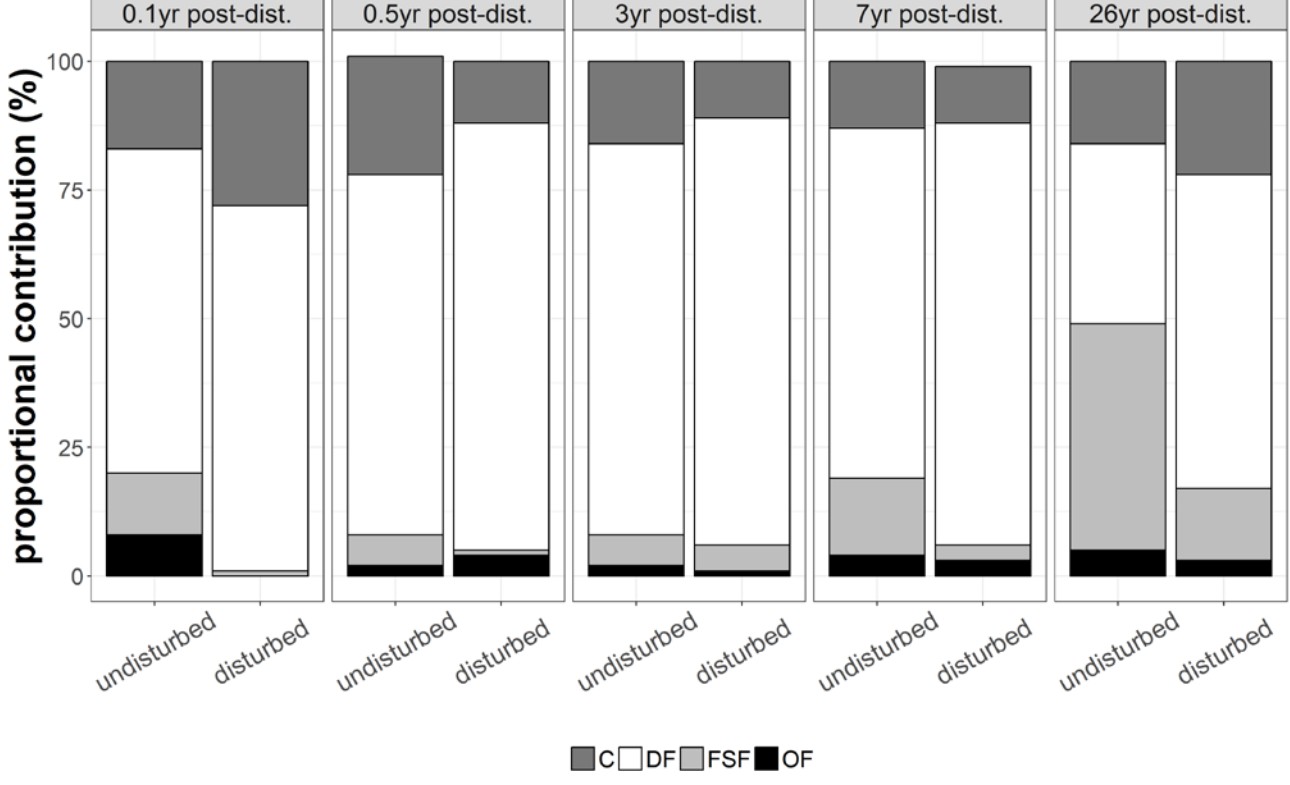

**Figure 3.** Proportional contribution (in %) of the feeding types C = carnivores, DF = deposit feeders, FSF = filter and suspension feeders, OF = omnivores to the total biomass for the undisturbed and disturbed sites inside the DISCOL experimental area (Peru Basin, SE Pacific) 0.1 years post-disturbance ($PD_{0.1}$), for 0.5 years post-disturbance ($PD_{0.5}$), for 3 years post-disturbance ($PD_3$), for 7 years post-disturbance ($PD_7$) and for 26 years post-disturbance ($PD_{26}$).





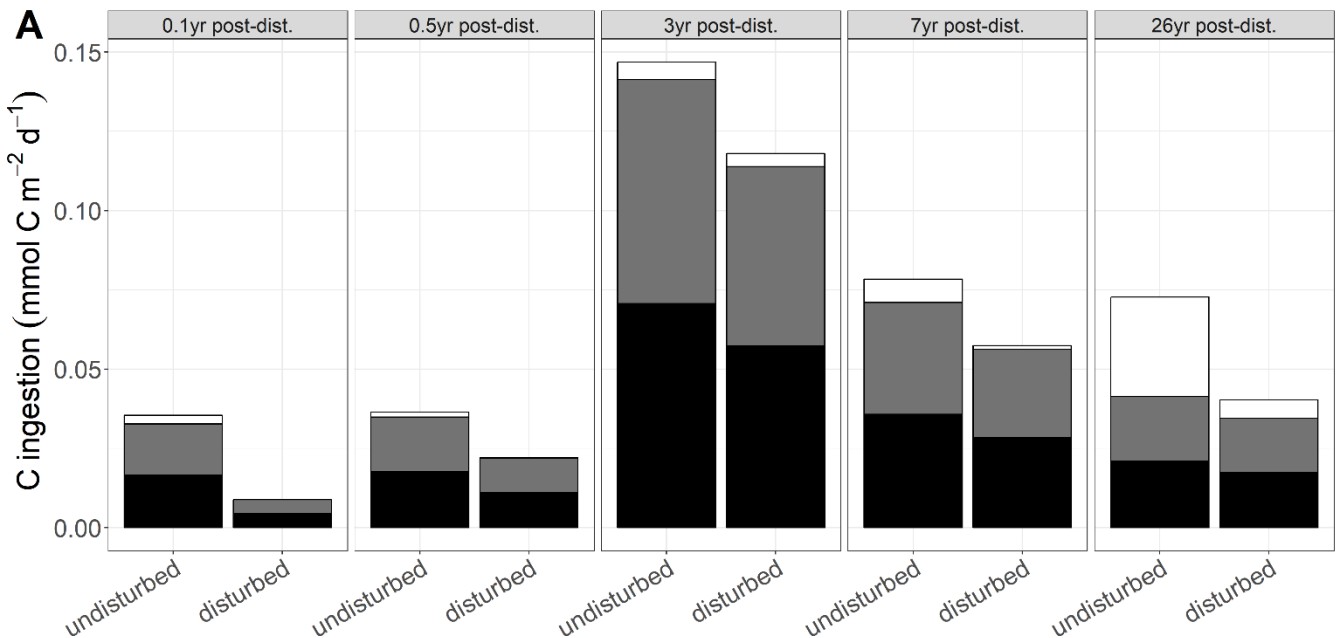

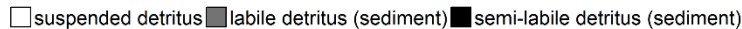

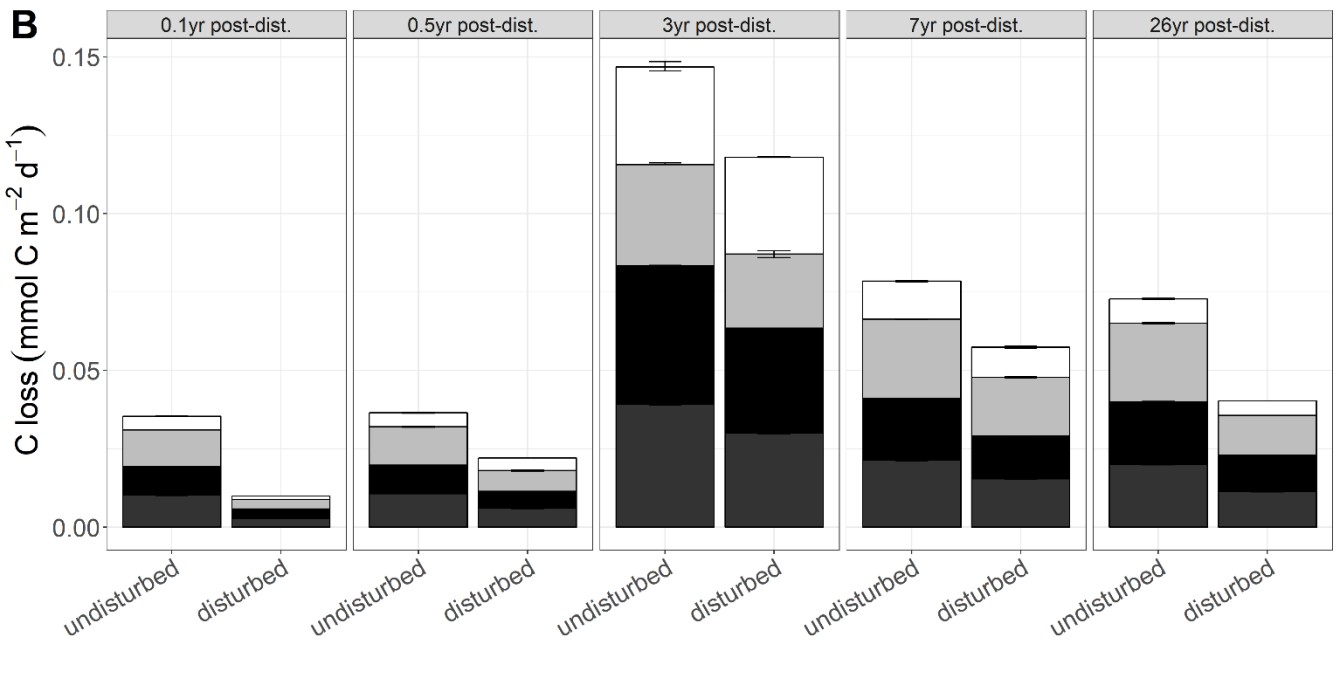





**Figure 4**. A) Mean faunal carbon ingestion (mmol C m$^{-2}$ d$^{-1}$) as suspended detritus, sedimentary labile and sedimentary semi-labile detritus for the undisturbed and disturbed sites the DISCOL experimental area (Peru Basin, SE Pacific) 0.1 years post-disturbance (PD$_{0.1}$), 0.5 years post-disturbance (PD$_{0.5}$), 3 years post-disturbance (PD$_3$), 7 years post-disturbance (PD$_7$) and 26 years post-disturbance (PD$_{26}$). B) Mean carbon losses (mmol C m$^{-2}$ d$^{-1}$) from the food webs as predation, faeces, scavenging on the carcass, and faunal respiration for the undisturbed and disturbed sites at PD$_{0.1}$, PD$_{0.5}$, PD$_3$, PD$_7$ PD$_{26}$. In both figures, the error bars represent 1 standard deviation.





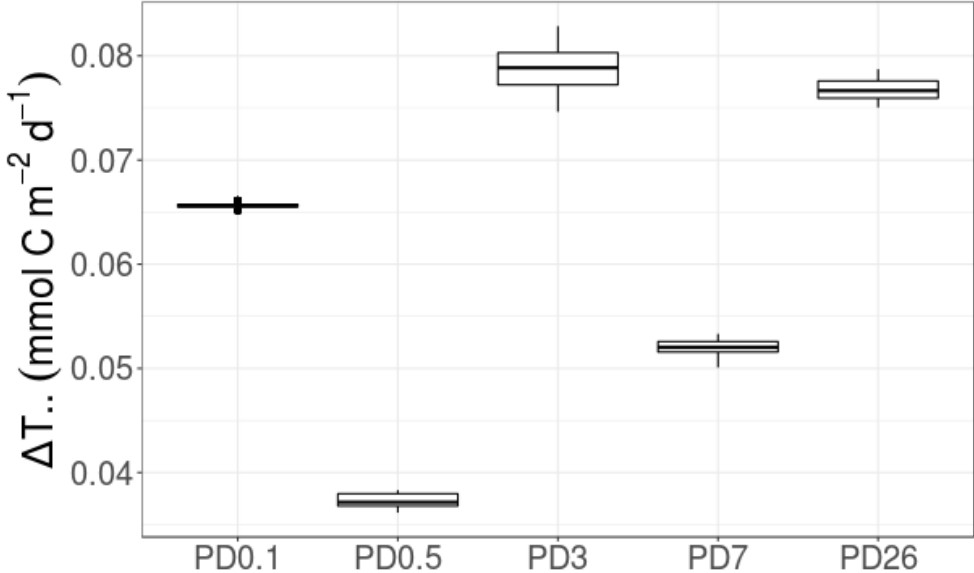

**Figure 5.** Development of $\Delta T..$ (mmol C m$^{-2}$ d$^{-1}$), i.e. the difference in 'total system throughput' $T..$ from the undisturbed compared to the disturbed sites, over time. $PD_{0.1}$ corresponds to 0.1 years post-disturbance, $PD_{0.5}$ is 0.5 years post-disturbance, $PD_3$ is 3 years post-disturbance, $PD_7$ is 7 years post-disturbance and $PD_{26}$ is 26 years post-disturbance.





**Figure 6.** Feeding-type related differences in the recovery of faunal respiration (mmol C m$^{-2}$ d$^{-1}$) over time following the
DISCOL disturbance experiment. Due to a lack of pre-disturbance respiration rates (T$_0$), the respiration rate for each feeding
type (filter and suspension feeders=FSF, surface deposit feeders=SDF, subsurface deposit feeders=SSDF, fish) is standardized
to the respective feeding type specific respiration rate at the undisturbed sediment of 0.1 years post-disturbance. The respiration
rate for filter and suspension feeders includes the respiration of macrofaunal, polychaete and megafaunal filter and suspension





feeders. The surface deposit feeders are the polychaete surface deposit feeders and the subsurface deposit feeders correspond

to the polychaete subsurface deposit feeders. Fish are the scavengers and predators.

