# Peer review of "Abyssal plain faunal carbon flows remain depressed 26 years after a simulated deep-sea mining disturbance"

_Biogeosciences, 2018_

## Referee Comment (RC1) · P. Jumars (Referee) · 17 Apr 2018

P. Jumars (Referee)

jumars@maine.edu

This manuscript represents a substantial contribution in the form of unique, long-term, experimental data on effects of disturbance on deep-sea communities from mining activities. The approach and methods appear sound. The sole apparent substantive exception to high quality of the presentation is the reporting of precision in Tables 1 and 2. Up to six significant figures are given (for Ceriantharia in Table 1), and several of the standard deviations include biomasses below zero. In general, precision for means and deviations should be comparable and should exclude the impossible. The authors

seem to have defaulted to an arbitrary two places after an arbitrarily placed decimal point. In a more minor but related issue, in Fig. 1 the color scheme makes the error bars very hard to discern.

The approach used to estimate individual biomass of Bryozoa and Hemichordata seems shaky enough that I would recommend doing the calculations with and without those estimates to convince myself that the results are not overly sensitive to their inclusion. Most Bryozoans are colonial, making me wonder what this individual biomass means.

---

## Referee Comment (RC2) · P. Jumars (Referee) · 20 Apr 2018

[revised manuscript text omitted]

Comment [PA2]: Tackling a hurdle is a mixed metaphor (U.S. football and track). Nobody intentionally tackles hurdles.

Measuring the carbon content of a macrofaunal specimen requires its complete combustion, which means that the specimen cannot be kept as a voucher . The acrofaunal samples collected for this study are part of the Biological Research Collection of Marine Invertebrates (Department of Biology & Centre for Environmental and Marine Studies, University of Aveiro, Portugal) and were therefore not sacrificed. Instead, we used the C conversion factors of macrofaunal specimens previously collected within the framework of a pulse-chase experiment in the Clarion-Clipperton

Zone (CCZ, NE Pacific), in which a deep-sea benthic lander (3 incubation chambers à $20 \times 20 \times 20$ cm) was deployed at water depths between 4050 and 4200 m (Sweetman et al., in review). The upper 5 cm of the sediment of the incubation chambers was were sieved on a 300 μm sieve and preserved in 4% buffered formaldehyde solution. Ashore, the samples were sorted and identified under a dissecting microscope, and the biomass of individual freeze-dried, acidified specimens was determined with at Thermo Flash EA 1112 elemental analyser (EA; Thermo Fisher Scientific, USA) to give the individual carbon contents in mmol C ind$^{-1}$. The mMacrofaunal density data (ind. m$^{-2}$) from all cruises were converted to macrofaunal biomass (mmol C m$^{-2}$) by multiplying each taxon-specific density (ind. m$^{-2}$) with the mean, taxon-specific, individual biomass value for macrofauna (mmol C ind$^{-1}$;

**Table 1**). Subsequently, the biomass data of all taxa with the same feeding type (

**Table 1**) were summed to calculate the biomass of each macrofaunal compartment (mmol C m$^{-2}$; Supplement 1, Figure 1).

The megafaunal density data (ind. m$^{-2}$) of the time series was converted to biomass (mmol C m$^{-2}$) by multiplying the taxon-specific density with a taxon-specific mean biomass per megafaunal specimen (mmol C ind$^{-1}$;

**Table 1**). To determine this taxon-specific biomass per megafaunal specimen, size measurements were used as follows. The 'AUV Abyss' (Geomar Kiel) equipped with a Canon EOS 6D camera system with 8-15 mm f4 fisheye zoom lens and 24 LED arrays for lightning (Kwasnitschka et al., 2016) flew approximately 4.5 m above the seafloor at a speed of 1.5 m s$^{-1}$ and took one picture every second (Greinert, 2015). Machine-vision processing was used to generate a photo-mosaic (Kwasnitschka et al., 2016). A subsample covering an area of 16,206 m$^2$ of the mosaic was annotated using the web-based annotation software 'BIIGLE 2.0' (Langenkämper et al., 2017). Lengths of all megafaunal taxa  for which data were available from previous cruises  were measured using the approach presented in Durden et al. (2016). Briefly, depending on the taxon, either body length, the diameter of the disk, or the length of an arm  was measured on the photo-mosaic and converted into biomass per individual (g ind$^{-1}$) using the relationship between measured body dimensions (mm) and preserved wet weight (g ind$^{-1}$) (Durden et al.,

2016). Subsequently, the preserved wet weight (g ind$^{-1}$) was converted to fresh wet weight (g ind$^{-1}$) using conversion factors from Durden et al. (2016) and to organic carbon (g C ind$^{-1}$ and mmol C ind$^{-1}$) using the taxon-specific conversion factors presented in Rowe (1983). For the taxa Cnidaria and Porifera no conversion factors were available. Therefore, taxon-specific individual biomass values were extracted from a study from the CCZ (Tilot, 1992). The individual biomass of Bryozoa and Hemichordata were calculated as the average biomass of an individual deep-sea megafaunal organism (B, mmol C ind$^{-1}$) at

4100 m depth following from the ratio of the regression for total biomass and abundance by Rex et al. (2006):

$$BB = \frac{10^{(-0.734-0.00039\times depth)}}{10^{(-0.245-0.00037\times depth)}} \qquad (1)$$

Following the approach applied to the macrofaunal data set, individual biomasses of taxa with similar feeding types (

> **Comment [PA3]:** I read that you estimated the mass of an individual chordate and a colony of bryozoans to be the same. This seems pretty shaky.

[revised manuscript text omitted]

* * *
**Comment [PA5]:** Space on both sides of operators, equal signs and inequality signs.

**Comment [PA6]:** I doubt that the second place on the right of the decimal place is justified. Please assess and give only one decimal place if the second cannot be supported.

the total faunal biomass at the undisturbed sites, whereas at PD$_3$ the total faunal biomass at the disturbed sites was 71% of the total faunal biomass at the undisturbed sites. At PD$_{26}$, the faunal biomass at the disturbed sites was 54% of the biomass at the undisturbed sites. The absolute weighted Hedges' *d*  |d$_+$| of all faunal compartment biomasses for PD$_{0.1}$ to PD$_7$ ranged from 0.053±0.019 at PD$_{0.5}$ to 0.075±0.019 (Supplement 2), indicating a strong experimental effect and therefore that biomasses of all faunal compartment failed to recover over the period analysed (PD$_{0.1}$ to PD$_7$).

The faunal biomass at both the undisturbed and disturbed sites from PD$_{0.1}$ to PD$_7$ was dominated by deposit feeders (from 63% at undisturbed PD$_{0.1}$ to 83% at disturbed PD$_{0.5}$ and disturbed PD3) (Figure 3). In contrast, at the undisturbed sites of PD$_{26}$, the largest contribution to total faunal biomass was from filter- and suspension feeders (44%), whereas deposit feeders only contributed 35%. At the disturbed sites of PD$_{26}$, deposit feeders had the highest biomass (61%), followed by carnivores (19%)

and filter- and suspension feeders (14%).

**3.2 Carbon flows**

total faunal C ingestion (mmol C m$^{-2}$ d$^{-1}$) ranged from 8.63×10$^{-3}$±1.58×10$^{-5}$ at the disturbed sites at PD$_{0.1}$ to 1.47×10$^{-1}$±8.55×10$^{-4}$ at the undisturbed sites at PD$_3$ and was always lower at the disturbed sites compared to the undisturbed sites (Figure 4A; Supplement 3). The ingestion consisted mainly of  sedimentary detritus (labile and semi-labile) that contributed between 56.97% (undisturbed sites, PD$_{26}$) and 99.50% (disturbed sites, PD$_{0.1}$) to  total carbon ingestion.

Faunal respiration (mmol C m$^{-2}$ d$^{-1}$) ranged from 6.02×10$^{-3}$±6.75×10$^{-5}$ (disturbed sites, PD$_{0.5}$) to 3.92×10$^{-2}$±3.69×10$^{-4}$ (undisturbed sites, PD$_3$). During the 26 yrears after the DISCOL experiment, modelled faunal respiration was always higher at undisturbed sites than at disturbed sites (Table 2, Figure 4). Over time, non-polychaete macrofauna contributed least to total faunal respiration (Table 2), except at the disturbed sites of PD$_{0.5}$ and at both sites of PD$_3$. During this

PD$_3$ sampling campaign, macrofauna contributed 49.97% at the undisturbed sites and 58.35% at the disturbed sites to  total faunal respiration. Polychaetes respired between 18.59% of  total fauna**l** respiration at the undisturbed sites at PD$_{26}$ and 77.61% of  total fauna**l** respiration at the disturbed sites at PD$_{0.5}$. The egafauna**l**  contribution to respiration was highest at PD$_{26}$,  when they respired 64.95% of the total faunal respiration at the disturbed sites and 78.67% of the total faunal respiration at the undisturbed sites. The contribution of fish to total faunal respiration was always <2%. Besides respiration, faeces production contributed between 20.07% at disturbed PD$_3$ and 34.65% at disturbed PD$_{0.1}$ to total carbon outflow from the food web (Figure 4). The contribution of the combined outflow of predation by external predators and scavengers on carcasses to the total C loss from the food web ranged from 50.48% at disturbed PD$_7$ to 65.33% at disturbed PD$_{0.1}$.

The fraction of *T*.. values that were larger for the food webs at the undisturbed sites than for the disturbed sites from the same sampling event was 1.0 at PD$_{0.1}$, PD$_{0.5}$, PD$_3$, PD$_7$ and PD$_{26}$. No decreasing trend in $\Delta T$.. over time was visible (Figure 5): in

**Comment [PA7]:** How many decimal places are warranted in your percentages. I suspect it is either none or one past the decimal point.

[revised manuscript text omitted]

[a]Taxon-specific individual biomass; [b]Individual biomass calculated based on all other macrofauna data; [c]Median taxon-specific individual biomass for individuals from the Porcupine Abyssal Plain where Durden et al. (2016) did not have reliable dimension measurements; [d]Mean taxon-specific biomass data per individual were extracted from Tilot (1992) for the CCZ; [e]Individual biomass of *Benthodytes* sp., one of the most abundant holothurian morphotype at the DISCOL site (Stratmann et al., in review); [f]Individual biomass of *Ipnops* sp., the most abundant deep-sea fish at the $PD_{26}$ undisturbed site; [g]Individual biomass calculated for mean  benthic megafauna at 4100 m depth based on the biomass-bathymetry and abundance-bathymetry relationships presented in Rex et al. (2006).

**Table 2.** Faunal respiration rate (mmol C m$^{-2}$ d$^{-1}$) and contribution (%) of the size classes macrofauna, polychaetes, megafauna and fish to the respiration for the undisturbed (Undist.) and disturbed (Dist.) sites directly after the disturbance event in March 1989 (PD$_{0.1}$), 0.5 years post-disturbance (September 1989, PD$_{0.5}$), 3 years post-disturbance (January 1992, PD$_3$), 7 years post-disturbance (February 1996, PD$_7$) and 26 years post-disturbance (September 2015, PD$_{26}$).

| | PD$_{0.1}$, Undist. | PD$_{0.1}$, Dist. | PD$_{0.5}$, Undist. | PD$_{0.5}$, Dist. | PD$_3$, Undist. | PD3, Dist. | PD7, Undist. | PD7, Dist. | PD26, Undist. | PD26, Dist. |
|---|---|---|---|---|---|---|---|---|---|---|
| Faunal respiration | $1.02\times10^{-2}\pm1.17\times10^{-4}$ | $2.72\times10^{-3}\pm5.23\times10^{-6}$ | $1.07\times10^{-2}\pm5.73\times10^{-5}$ | $6.02\times10^{-3}\pm6.75\times10^{-5}$ | $3.92\times10^{-2}\pm3.68\times10^{-4}$ | $2.99\times10^{-2}\pm2.33\times10^{-4}$ | $2.14\times10^{-2}\pm2.50\times10^{-4}$ | $1.54\times10^{-2}\pm1.49\times10^{-4}$ | $2.00\times10^{-2}\pm1.50\times10^{-4}$ | $1.13\times10^{-2}\pm1.04\times10^{-4}$ |
| Macrofauna | 8.63 | 7.34 | 9.73 | 14.35 | 49.97 | 58.35 | 6.50 | 4.51 | 2.64 | 1.19 |
| Polychaeta | 61.59 | 77.80 | 62.69 | 77.61 | 27.09 | 30.03 | 67.08 | 83.51 | 18.52 | 32.43 |
| Megafauna | 29.47 | 14.85 | 27.06 | 8.04 | 22.30 | 11.54 | 25.75 | 11.63 | 78.67 | 64.95 |
| Fish | $3.02\times10^{-1}$ | 0.00 | $5.29\times10^{-1}$ | 0.00 | $6.43\times10^{-1}$ | $7.75\times10^{-2}$ | $6.64\times10^{-1}$ | $3.53\times10^{-1}$ | $1.73\times10^{-1}$ | 1.44 |

**Comment [MOU8]:** Check precision

[Figure]

**Comment [MOU9]:** Change your color scheme. Your error bars are effectively invisible on many of the bar plots

**Figure 1.** Mean biomass (mmol C m$^{-2}$) of the  food-web compartments for the undisturbed and disturbed sites inside the DISCOL experimental area (Peru Basin, SE Pacific) 0.1 years post -disturbance (PD$_{0.1}$), for 0.5 years post -disturbance (PD$_{0.5}$), for three years post -disturbance (PD$_3$), for seven years post -disturbance (PD$_7$), and for 26 years post -disturbance (PD$_{26}$). The error bars represent 1 standard deviation.

The abbreviation are: MacC = macrofauna carnivores, MacDF = macrofauna deposit feeders, MacFSF = macrofauna filter/ suspension feeders, MacO = macrofauna omnivores, MegC = megafauna carnivores, MegDF = megafauna deposit feeders, MegFSF = megafauna filter/ suspension feeders, MegOF = megafauna omnivores, PolC = polychaete carnivores, PolOF = polychaete omnivores, PolSDF = polychaete surface deposit feeders, PolSF = polychaete suspension feeders, PolSSDF =

polychaete subsurface deposit feeders.

[Figure]

**Figure 2.** Simplified schematic representation of the  food-web structure that forms the basis of the linear inverse model (LIM). All compartments inside the box were part of the  food-web model, whereas compartments outside the black box were only considered as carbon influx or efflux, but were not directly modelled. In order to simplify the graph, for macrofauna, polychaetes and megafauna, only feeding types were presented and no size classes. Solid black arrows represent the carbon flux between food-web compartments and black dashed arrows represent the influx of carbon to the model. Blue-dotted arrows show the loss of carbon from the food web via respiration to DIC. The  red-dashed arrows indicate the loss of carbon from the food web as faeces and as predation by pelagic/ benthopelagic fish and the yellow-dashed arrow indicate the reduction of the carcass pool due to scavenging by pelagic/ benthopelagic fish.

[Figure]

**Figure 3.** Proportional contribution (in %) of the feeding types C = carnivores, DF = deposit feeders, FSF = filter and suspension feeders, OF = omnivores to the total biomass for the undisturbed and disturbed sites inside the DISCOL experimental area (Peru Basin, SE Pacific) 0.1 years post-disturbance ($PD_{0.1}$), for 0.5 years post-disturbance ($PD_{0.5}$), for 3 years post-disturbance ($PD_3$), for 7 years post-disturbance ($PD_7$) and for 26 years post-disturbance ($PD_{26}$).

**Comment [MOU10]:** Use the abbreviation for years (yr) after Arabic numerals, both in this caption and the others.

**A**

[Figure]

□ suspended detritus  ■ labile detritus (sediment)  ■ semi-labile detritus (sediment)

**B**

[Figure]

□ predation  ■ feces  ■ scavenging  ■ respiration

**Figure 4**. A) Mean faunal carbon ingestion (mmol C m$^{-2}$ d$^{-1}$) as suspended detritus, sedimentary labile and sedimentary semi-labile detritus for the undisturbed and disturbed sites the DISCOL experimental area (Peru Basin, SE Pacific) 0.1 years post-disturbance (PD$_{0.1}$), 0.5 years post-disturbance (PD$_{0.5}$), 3 years post-disturbance (PD$_3$), 7 years post-disturbance (PD$_7$) and 26 years post-disturbance (PD$_{26}$). B) Mean carbon losses (mmol C m$^{-2}$ d$^{-1}$) from the food webs as predation, faeces, scavenging on the carcass, and faunal respiration for the undisturbed and disturbed sites at PD$_{0.1}$, PD$_{0.5}$, PD$_3$, PD$_7$ PD$_{26}$. In both figures, the error bars represent 1 standard deviation.

[Figure]

**Figure 5.** Development of $\Delta T..$ (mmol C m$^{-2}$ d$^{-1}$), i.e., the difference in 'total system throughput' $T..$ from the undisturbed compared to the disturbed sites, over time. $PD_{0.1}$ corresponds to 0.1 years post-disturbance, $PD_{0.5}$ is 0.5 years post-disturbance, $PD_3$ is 3 years post-disturbance, $PD_7$ is 7 years post-disturbance and $PD_{26}$ is 26 years post-disturbance.

[Figure]

**Figure 6.** Feeding  type-related differences in the recovery of faunal respiration (mmol C m$^{-2}$ d$^{-1}$) over time following the DISCOL disturbance experiment. Due to a lack of pre-disturbance respiration rates (T$_0$), the respiration rate for each feeding type (filter and suspension feeders=FSF, surface deposit feeders=SDF, subsurface deposit feeders=SSDF, fish) is standardized to the respective feeding type specific respiration rate at the undisturbed sediment of 0.1 years post-disturbance. The respiration rate for filter and suspension feeders includes the respiration of macrofaunal, polychaete and megafaunal filter and suspension feeders. The surface deposit feeders are the polychaete surface deposit feeders and the subsurface deposit feeders correspond to the polychaete subsurface deposit feeders. Fish are the scavengers and predators.

---

## Referee Comment (RC3) · Anonymous Referee #2 · 20 Apr 2018

General Comments:

This manuscript represents an important contribution to scientific knowledge of the possible impacts of seabed mining on deep-sea communities. The work conducted is original and timely, and the aims of the manuscript are clear and focussed. Analyses conducted are generally of a high standard, and the quality of figures and tables is satisfactory.

However, the manuscript could be strengthened by some additional simple analyses

(detailed in the 'Specific Comments' section), as well as greater recognition of the limitations of thee data sets analysed.

In particular, I feel the naming of sites within the DISCOL experimental area that were not directly ploughed as 'undisturbed' misleading. Although not ploughed, such sites will still likely have experienced disturbance in the form of settlement of re-suspended sediments; a projected impact of deep-sea mining noted by the authors on page 2, line 26 of the manuscript. This is an issue which should be discussed in the manuscript but is not presently recognised.

Related to this, the authors describe the DISCOL experiment as a 'simulated small-scale deep-sea mining experimental disturbance' (Page 2 lines 7-8). However, no nodules were removed during the DISCOL experiment, and in this way, amongst others, DISCOL was not a perfect simulation of disturbance caused by deep-sea mining. How the results of this study may differ if nodules are removed from the sediment deserves discussion.

A glaring omission to the LIM analysed in this manuscript is the lack of microbial and meiofaunal data. It is explained in the manuscript that this is because of insufficient data (page 3, lines 10-11). This is understandable, but the impact this lack of microbial and meiofaunal data may have had on the analyses conducted deserves discussion in the 'Model limitations' section of the manuscript.

Another limitation of this study is the lack of baseline sampling – a 'Pre-Disturbance' time point. Such a time point would give a better indication of the 'undisturbed' ecosystem state against which all post-disturbance time points could be compared (especially PD0.1). Clearly it is not possible to obtain this data now, but this lack of baseline data requires discussion. At present it is only noted in the legend of figure 6.

Throughout the manuscript, the high values of community metrics obtained for the PD3 time point are noted frequently – highest biomass, highest faunal C ingestion, highest respiration, highest macrofaunal contribution, lowest faeces contribution to total C outflow etc. However, no attempt is made to explain this observation. Similarly, on multiple occasions, 'natural variability' is noted amongst observations. I find it surprising that no attempt has been made to identify the variable(s) that may be driving this variability, even qualitatively. If quantitative analyses to this effect are not possible, the manuscript would still benefit from greater discussion of this natural variability.

Whilst the use of Jumars' (1981) paper to structure the discussion of this manuscript is an excellent idea, I feel that the quality of this discussion could be improved. For example, on page 11, line 28 it is stated that "...Jumars' (1981) predictions for sub-surface deposit feeders could not be tested...". In the 'Specific Comments' section, I have given some suggestions of simple analyses which could be used to test Jumars' predictions more effectively.

Finally, whilst the quality of language used in this manuscript is satisfactory, many sentences, particularly in the discussion section could be re-written to improve the flow of the manuscript (e.g. the first sentence of section 4.1). There are also a number of grammatical errors, but I have not detailed these in full.

Specific comments:

Title

The title is rather long and could be improved to increase the impact of the manuscript. I suggest something like 'Abyssal plain faunal carbon flows remain depressed 26 years after a simulated deep-sea mining disturbance'.

Abstract

It is not immediately clear upon a first read-through of the abstract that LIM were produced for all points in the time-series, rather than just at PD26 (as the title may suggest). Please make this clearer.

Percentages – is it possible to give some sense of variability around these values?

At nearly 400 words, the abstract would be improved by more a concise wording.

Introduction

The description of the DISCOL experiment (page 3, paragraph 1) could be clearer. Perhaps a figure illustrating the areas ploughed/not ploughed would help. Also, I feel it would be better for only a short introduction to the DISCOL experiment to be included in the Introduction section, with a more detailed description being reserved for the 'Methods' section.

Page 3, paragraph 2: It is good to introduce the basics of LIM in the Introduction section. However, the current text is perhaps a little too technical for this section. I suggest moving the more technical aspects of this paragraph to a new paragraph in the Methods section.

I would like to see the aims of this study, and perhaps some hypotheses (e.g. based on Jumars (1981)), stated more explicitly at the end of this section.

Methods

Page 4, paragraph 1: It would be useful for the reader to know how many box cores were collected from 'disturbed' and 'undisturbed' sites at every time point, not just the PD26. This could be easily detailed in a table, which could also detail surface area of seabed surveyed for megafauna. Alternatively, the locations of all box cores collected over the 26-year study period could be plotted (colour-coded for date of collection) on the figure I proposed above illustrating the DISCOL experimental mining disturbance regime.

Page 4, line 8: Only three box cores were collected from disturbed sites for PD26. This is a very low level of replication, and something which should be discussed in the 'Model Limitations' section.

Page 4, paragraph 1: It seems strange that such an effort was made to analyse the same number of images for 'disturbed' and 'undisturbed' sites for megafauna, but there

was no corresponding effort to analyse the same number of box cores for 'disturbed' and 'undisturbed' sites. Was an effort made to standardise megafaunal sampling effort for the other post-disturbance time points to that of PD26?

Page 4/5: Conversion of biomass into carbon content; I would like to know exactly what the conversions used were, if possible. These could be included in a supplementary materials file.

Page 5, line 20: No details are given of the conversion used for cnidarian/ poriferan biomass to carbon content. The authors should elaborate on the use of the Tilot (1992) paper.

The section detailing biomass to carbon content conversion would be made clearer by greater consistency in the use of the term 'biomass' to mean either the total weight of individuals, or the total carbon weight of individuals.

Page 6, lines 4-9: What literature was used to determine the coarse feedings guilds assigned to other taxa?

Page 6, lines 7: It is stated that "...a further detailed classification of the macrofaunal polychaetes..." was made. However, this further detailed classification seems only to additionally subdivide deposit feeding polychaetes into surface/subsurface categories. Could this division not be made with a little effort for all invertebrate macrofauna and megafauna?

Page 7 line 23 to page 8 line 9: Why was 'Hedge's d' used here rather than t-tests or their non-parametric equivalent? This should be clarified.

Results

Page 8, lines 18-21: It is interesting that the minimum and maximum biomass values occur at same PD time steps for both 'disturbed' and 'undisturbed' treatments. That the lowest biomass for 'undisturbed' sites was observed at PD0.1 may suggest that these sites were actually disturbed to an extent. The fact that both disturbance treatments

reach maximal biomass at the same PD time point may also suggest a shared recovery trajectory following disturbance.

Page 8, lines 21-23: This comparison of the change in % biomass difference between disturbance treatments from PD0.1 to PD3 is verging on discussion. I suggest moving it to the Discussion section.

Page 8, lines 24-26: Why is 'absolute weighted Hedge's d |d+|' reported here when on page 8, lines 6-9, Hedge's d is explained in a different form? This is confusing for the reader as page 8 lines 6-9 suggests that values greater than ∼0.8 represent strong effect sizes, but the results given on page 8 lines 24-26 report small values associated with the metric and the authors describe these as 'indicating a strong experimental effect'.

Discussion

Page 9, line 28: Suggest changing "...compared to the undisturbed sediment after 26 years" to "...compared to the undisturbed sediment 26 years after experimental mining disturbance".

Page 10, lines 1-23: I feel the authors are somewhat underselling the conclusions of their manuscript by placing a model limitations section so early on in their Discussion. This could be moved to later in the manuscript, perhaps to just before the conclusions.

Page 11, line 7: I am confused why the authors are discussing changes in fish respiration over long time periods (3 years) at undisturbed sites. The predictions of Jumars (1981) would be better tested by considering changes in respiration at disturbed sites very soon after the disturbance (e.g. PD0.1). I note that no fish were detected at disturbed sites at PD0.1, so simply put, this hypothesis cannot be tested with this data set.

Page 11, line 28: "Hence, Jumars (1981) predictions for sub-surface deposit feeders could not be tested...". Indeed, it would be easier to test Jumars' predictions if a PD0

time point was available. However, would it not be possible to test whether there is a significant difference in the density of subsurface deposit feeders at PD0.1 between the disturbance categories? Under Jumars' predictions, we would expect the density of sub-surface deposit feeders to be much reduced at 'disturbed' sites relative to 'undisturbed' sites at this time point.

Page 11/12, lines 30-6: The authors state Jumars' prediction that surface deposit feeders will be more drastically impacted by mining activities than sub-surface deposit feeders. However, they do not test this prediction, instead comparing deposit feeder ecosystem functioning to that of 'omnivores, filter- and suspension feeders and carnivores'. Please explain why. It would be possible to investigate the relative changes in surface and sub-surface deposit feeder contributions to ecosystem functioning between 'disturbed' and 'undisturbed' sites.

Page 12, lines 15-16: "After 26 years, the relative difference in the filter and suspension feeding respiration rate was still 80%". I assume that this refers to the difference in respiration rate of filter and suspension feeders between the disturbance categories? The current text is ambiguous and could be interpreted as the difference in respiration rate between filter and suspension feeders. It is also unclear whether an 80% difference means that respiration rates at 'disturbed' sites were 80% lower than at 'undisturbed', or that respiration rates at 'disturbed' sites were 80% of those at 'undisturbed'.

Page 12, lines 18-19: "...indicating a slow recovery of this feeding group". I'd argue that compared to Jumars' predictions this apparent recovery rate is relatively fast!

Page 12, lines 23-24: The authors complain here and elsewhere about natural variability in values making it difficult to isolate disturbance-related trends. However, the authors make no effort to identify or even simply discuss the key environmental factors which may be driving this variability.

Page 12, lines 20-28: This summary paragraph is unnecessary.

Page 13, lines 5-6: "In contrast, filter and suspension feeders did not recover at all...". This sentence is too strongly worded. The authors state on page 12, lines 13-15, that "Directly after the initial DISCOL disturbance event, the respiration rate of filter and suspension feeders at the disturbed sediment was only 1% of the respiration rate of this feeding type at the undisturbed sediment" and on page 12, lines 15-16, that "After 26 years, the relative difference in the filter and suspension feeding respiration rate was still 80%". Whilst I agree that the respiration rate of filter- and suspension feeders is still clearly depressed at 'disturbed' sites relative to 'undisturbed', even 26 years post-disturbance, there clearly has been some recovery - perhaps even more so than might be expected! Please re-word this sentence to soften your conclusions.

Page 13, line 7: The authors state that "...[ecosystem functioning] has not recovered 26 years after the experimental disturbance". However, there is clearly some evidence of recovery. Please could the authors change this statement to "...[ecosystem functioning] has not fully recovered 26 years after the experimental disturbance"?

Table 1

Please explain what 'n' stands for. Is this the number of taxa analysed, or the number of individuals used to estimate taxon-specific biomass etc.?

Figure 1

There is a lot of information on this figure, and the overlap in error bars make it especially difficult to read. One option would be to plot each group separately, although this would result in a large number of graphs. Alternatively, this information may be more clearly presented as a table (as per Table 2). Why are there no error bars for the PD26 bars?

Figure 4

Why are there no error bars on figure 4a? Are they simply too small to see?

Technical corrections:

Abstract

Page 1, line 27: I, and most others, consider fish as megafauna. Please explain why they are treated separately to the other megafauna.

Introduction

Page 2, line 12: The word 'occasionally' is used twice in same sentence. Page 2, line 20: Yttrium is typically considered a rare earth element. Page 2, line 25: One could argue that there's not really such a thing as food-rich surface sediments on the deep seafloor. Page 3, line 2: '10.8 km2 large circular area' – 'large' is not required.

Methods

Page 4, line 21: 'could' should be used, not 'can'.

Discussion

Page 9, line 23: Suggest changing 'evolution' for 'change over time'. Page 9, line 27: Put in comma after "...role of the various feeding types in the carbon cycling differs", and after "...was significantly lower". Page 11, line 28: "Hence, Jumars (1981) predictions..." should be 'Hence, Jumars' (1981) predictions...'. Page 11, line 34: "...deposit feeders seem to have advantages during the recovery from the DISCOL disturbance experiment...". Relative to whom?

Figure 1

'Figure 1' is actually referred to in the text after 'figure 2' is. Swap around the order of these figures – i.e. 'figure 2' should be renamed 'figure 1', and vice versa.

Figure 2

Inconsistent spelling of faeces here and throughout the manuscript. Figure legend line 7 "...yellow-dashed arrow indicate..." should be '...yellow-dashed arrow indicates...'. Incorrect use of 'due to' here and throughout the manuscript. Please change to 'because

of' or 'as a result of'.

Figure 5

Is it possible to subscript the x-axis post-disturbance times – e.g. PD0.1, for consistency with the rest of the manuscript?

―――――――――――――――

---

## Author Comment (AC2) · 10 Jun 2018

All spelling and grammar mistakes that Peter Jumars corrected in the supplement were accepted in the revised manuscript.

---

## Author Response (AR1)

25 June 2017

Biogeosciences
Attn..: Matthias Haeckel, Special Issue Associated Editor

Dear Matthias Haeckel,

Please consider our manuscript now entitled "Abyssal plain faunal carbon flow remain depressed 26 years after a simulated deep-sea mining disturbance" for publication in the special issue "Assessing environmental impacts of deep-sea mining – revisiting decade-old benthic disturbances in Pacific nodule areas" of Biogeosciences.

We would like to thank you for handling our manuscript and are grateful for the positive and detailed feedback provided by Peter Jumars (Reviewer #1) and Reviewer #2 for our manuscript "Faunal carbon flows in the abyssal plain food web of the Peru Basin have not recovered during 26 years from an experimental sediment disturbance". The main issues identified by the reviewers considered reporting of precision, estimation of byrozoan biomasses, and naming of sampling sites. We addressed them by reporting all data with 3 significant figures in the text and only 1 decimal in tables. Bryozoan biomass estimates were removed from the table because they were mistakenly reported. In fact, bryozoans were only observed at references sites which were not modelled and not inside or outside plough tracks. We also changed the names to 'outside plough tracks' (previously 'undisturbed site') and 'inside plough tracks' (previously 'disturbed site'). We addressed your editorial comments by correcting the width of the plow harrow to 8 m, provided the PANGAEA DOI for the OFOS images and added the project short name in the acknowledgements. We added the biomass conversion factors mentioned in Tilot's PhD thesis in supplement 2 where we present all conversion factors used in the manuscript and provide the URL access to the digital copy of Tilot's PhD thesis in the references.

We addressed each of the comments of Peter Jumars and Reviewer #2 in detail below.

With these modifications, we hope that the manuscript is now suited for publication in Biogeosciences.

Looking forward to your decision and thank you again for handling our manuscript.

Kind regards,
Tanja Stratmann

Department of Estuarine and Delta Systems
NIOZ – Royal Netherlands Institute for Sea Research
Korringaweg 7
4401 NT Yerseke
The Netherlands

**Detailed responses**

**Reviewer #1**

**Reviewer 1 comment 1:** …the reporting of precision in Tables 1 and 2. Up to six significant figures are given (for Ceriantharia in Table 1), and several of the standard deviations include biomasses below zero. In general, precision for means and deviations should be comparable and should exclude the impossible. The authors seem to have defaulted to an arbitrary two places after an arbitrarily placed decimal point.
**Our response:** We adjusted the Result section 3.1, Table 1 and 2 and report all biomass data with a precision of 3 significant figures in the text and only one decimal place in Table 1. The individual biomasses of organisms in Table 1 are reported as mean ± standard error and percentages are presented as integers in the text and with one decimal place in Table 2. We also corrected the remainder of the manuscript in this respect.

**Reviewer 1 comment 2:** In a more minor but related issue, in Fig. 1 the color scheme makes the error bars very hard to discern.
**Our response:** Also Reviewer #2 (see below) indicated that the error bars are difficult to see. Hence, we decided to remove the error bars from the plot for better visibility and refer to Supplement 1 for the standard deviations.

**Reviewer 1 comment 3:** The approach used to estimate individual biomass of Bryozoa and Hemichordata seems shaky enough that I would recommend doing the calculations with and without those estimates to convince myself that the results are not overly sensitive to their inclusion. Most Bryozoans are colonial, making me wonder what this individual biomass means.
**Our response:** We mistakenly included bryozoans in Table 1 in our initial submission. In the study area bryozoans were only found in the reference sites and, as mentioned in the manuscript (Page 3 line 29), these sites were not modelled. Therefore, we removed bryozoans from Table 1 and from the rest of the manuscript.

All spelling and grammar mistakes that Peter Jumars corrected in the supplement were accepted in the revised manuscript.

General comments:

**General comment 1:** I feel the naming of sites within the DISCOL experimental area that were not directly ploughed as 'undisturbed' misleading. Although not ploughed, such sites will still likely have experienced disturbance in the form of settlement of re-suspended sediments; a projected impact of deep-sea mining noted by the authors on page 2, line 26 of the manuscript. This is an issue which should be discussed in the manuscript but is not presently recognised.

**Our response:** We agree with referee #2 that the naming we adopted from previous DISCOL publications is misleading and therefore changed it in the revised manuscript to 'inside plough tracks' which corresponds to the former 'disturbed sites' and 'outside plough tracks' corresponding to the previous 'undisturbed sites'.

**General comment 2:** The authors describe the DISCOL experiment as a 'simulated small-scale deep-sea mining experimental disturbance' (Page 2 lines 7-8). However, no nodules were removed during the DISCOL experiment, and in this way, amongst others, DISCOL was not a perfect simulation of disturbance caused by deep-sea mining. How the results of this study may differ if nodules are removed from the sediment deserves discussion.

**Our response:** For epifauna that are dependent on nodules as hard substrate there is no difference between removing the nodule or ploughing it into the sediment surface as in both cases the nodule disappears from the sediment surface. We therefore added the following two sentences: This hypothesis could not be tested directly, because nodules were not removed in this experiment, but only ploughed into the sediment. However, the disappearance of nodules from the sediment surface will have the same effect on sessile epifauna that depend on nodules as hard substrate independently of the method by which the nodules disappeared." Additionally, we describe more specifically which type of disturbance were created during the DISCOL experiment in the Introduction: "A 10.8 km$^2$ circular area (Figure 1) was ploughed diametrically 78 times with an 8 m wide plough-harrow; a treatment which did not remove nodules, but disturbed the surface sediment, buried nodules into the sediment and created a sediment plume (Thiel et al., 1989)."

**General comment 3:** A glaring omission to the LIM analysed in this manuscript is the lack of microbial and meiofaunal data. It is explained in the manuscript that this is because of insufficient data (page 3, lines 10-11). This is understandable, but the impact this lack of microbial and meiofaunal data may have had on the analyses conducted deserves discussion in the 'Model limitations' section of the manuscript.

**Our response:** We agree that the addition of microbial data would increase T.., the sum of carbon flows, but we do not know whether the overall trend (outside plough track vs. inside plough track) would change. Since we cannot adequately discuss this, but only speculate, we decided against a longer discussion in the model section. Furthermore, it is explicitly stated in the title that we modelled only faunal carbon flows.

**General comment 4:** Another limitation of this study is the lack of baseline sampling – a 'Pre-Disturbance' time point. Such a time point would give a better indication of the 'undisturbed' ecosystem state against which all post-disturbance time points could be compared (especially PD0.1). Clearly it is not possible to obtain this data now, but this lack of baseline data requires discussion. At present it is only noted in the legend of figure 6.

**Our response:** We briefly mention this in the introduction ("Therefore, the food-web models presented in this work cover post disturbance 1989 (no adequate pre-disturbance sampling

took place) to 2015 and contain only macrofauna, invertebrate megafauna and fish.") and discuss it in more detail in the 'Model limitation' section: "Pre-disturbance samples and samples from reference sites were not collected for all food-web compartments. We therefore lack a baseline to which the 'outside plough track' food web at $PD_{0.1}$ could be compared to assess the impact that the disturbance effect had on sites outside the plough tracks."

**General comment 5:** Throughout the manuscript, the high values of community metrics obtained for the PD3 time point are noted frequently – highest biomass, highest faunal C ingestion, highest respiration, highest macrofaunal contribution, lowest faeces contribution to total C outflow etc. However, no attempt is made to explain this observation. Similarly, on multiple occasions, 'natural variability' is noted amongst observations. I find it surprising that no attempt has been made to identify the variable(s) that may be driving this variability, even qualitatively. If quantitative analyses to this effect are not possible, the manuscript would still benefit from greater discussion of this natural variability.
**Our response:** The aim of our study was to test Jumars' predictions on ecosystem recovery after deep-sea mining with real data. We therefore do not consider our 'Feeding-type specific differences in recovery' section of the discussion an adequate place to discuss natural variability. However, we address this issue in the 'Model limitations' section now (page 10, lines 11-13): "We cannot identify either whether the high biomasses and as a result higher carbon flows at PD3 were correlated with the begin of the positive (La Niña) phase of the El Niño Southern Oscillation (Trenberth, 1997) which led to an abnormally high POC flux at Station M at the time of $PD_3$ (Ruhl et al., 2008)."

**General comment 6:** Whilst the use of Jumars' (1981) paper to structure the discussion of this manuscript is an excellent idea, I feel that the quality of this discussion could be improved. For example, on page 11, line 28 it is stated that "...Jumars' (1981) predictions for subsurface deposit feeders could not be tested...". In the 'Specific Comments' section, I have given some suggestions of simple analyses which could be used to test Jumars' predictions more effectively.
**Our response:** We appreciate these suggestions and address them below.

**General comment 7:** Finally, whilst the quality of language used in this manuscript is satisfactory, many sentences, particularly in the discussion section could be re-written to improve the flow of the manuscript (e.g. the first sentence of section 4.1).
**Our response:** We improved the manuscript using the detailed textual corrections from Peter Jumars and the later remarks by Reviewer #2. We also modified the first sentence of section 4.1.

Specific comments for the abstract:
**Specific comment 1:** The title is rather long and could be improved to increase the impact of the manuscript. I suggest something like 'Abyssal plain faunal carbon flows remain depressed 26 years after a simulated deep-sea mining disturbance'.
**Our response:** We accept the suggestion of the new title.

**Specific comment 2:** It is not immediately clear upon a first read-through of the abstract that LIM were produced for all points in the time-series, rather than just at PD26 (as the title may suggest). Please make this clearer.
**Our response:** We rephrased the sentence as follows: "We used this unique abyssal faunal time series to develop carbon-based food web models for each point in the time series using the linear inverse model (LIM) approach for sediments subjected to two disturbance levels: 1)

outside the plough tracks, not directly disturbed by plough, but probably suffered from additional sedimentation and 2) inside the plough tracks."

**Specific comment 3:** Percentages – is it possible to give some sense of variability around these values?
**Our response:** It would only be possible to present standard deviations or standard errors for flow values, but not for the biomass at $PD_{26}$, since no replicates are available for megafauna estimates. For consistency, we therefore decided not to report any variability around the percentages in the abstract. However, standard deviations or standard errors are reported in the main text and figures/ tables.

**Specific comment 4:** At nearly 400 words, the abstract would be improved by more a concise wording.
**Our response:** We shortened the abstract following the advice of both reviewers.

Specific comments for the introduction:
**Specific comment 5:** The description of the DISCOL experiment (page 3, paragraph 1) could be clearer. Perhaps a figure illustrating the areas ploughed/not ploughed would help. Also, I feel it would be better for only a short introduction to the DISCOL experiment to be included in the Introduction section, with a more detailed description being reserved for the 'Methods' section.
**Our response:** We added a figure showing the plough tracks inside the DISCOL experimental area, but we like to keep the focus of the 'Methods' section on the food-web models instead of describing the DISCOL experiment in more detail. Moreover, the DISCOL is described in detail in the referenced papers Thiel and Schriever (1989) and Bluhm (2001).

**Specific comment 6 (Page 3, paragraph 2):** It is good to introduce the basics of LIM in the Introduction section. However, the current text is perhaps a little too technical for this section. I suggest moving the more technical aspects of this paragraph to a new paragraph in the Methods section.
**Our response:** We moved the more technical parts of the paragraph to a new subsection in the method part and combined it the former Method section 2.4.

**Specific comment 7:** I would like to see the aims of this study, and perhaps some hypotheses (e.g. based on Jumars (1981)), stated more explicitly at the end of this section.
**Our response:** We rephrased this section and explicitly state three aims of our study.

Specific comments for the methods section:
**Specific comment 8:** It would be useful for the reader to know how many box cores were collected from 'disturbed' and 'undisturbed' sites at every time point, not just the PD26. This could be easily detailed in a table, which could also detail surface area of seabed surveyed for megafauna. Alternatively, the locations of all box cores collected over the 26-year study period could be plotted (colour-coded for date of collection) on the figure I proposed above illustrating the DISCOL experimental mining disturbance regime.
**Our response:** The requested data were already available in our original submission in Supplement 2, but we now also include them in Table 1. We decided, however, against plotting sampling stations on the map because several stations are taken so closely together that symbols would overlap considering the scale of the map.

**Specific comment 9 (Page 4, line 8):** Only three box cores were collected from disturbed sites for PD26. This is a very low level of replication, and something which should be discussed in the 'Model Limitations' section.

**Our response:** We understand this comment, but logistical reasons underlie this low replication. The box corer was not equipped with video guidance and could therefore not be positioned exactly on the 8-m plough tracks. As a result, only three of the boxcores hit the targeted tracks and could be allocated to the category "inside plough tracks". The remaining boxcores in the DISCOL experimental were conservatively assigned to "outside plough tracks". Owing to the ship time available for this type of sampling it was not possible to pursue with further attempts to hit the plough tracks.

**Specific comment 10 (Page 4, paragraph 1):** It seems strange that such an effort was made to analyse the same number of images for 'disturbed' and 'undisturbed' sites for megafauna, but there was no corresponding effort to analyse the same number of box cores for 'disturbed' and 'undisturbed' sites. Was an effort made to standardise megafaunal sampling effort for the other post-disturbance time points to that of PD26?

**Our response:** The low number of box core replicates was addressed above. We did not attempt to standardize megafaunal sampling to other post-disturbance time points because camera systems also differed between the cruises $PD_{0.1}$, $PD_{0.5}$, $PD_3$ and $PD_7$. Additionally, selective sampling which was used during the cruises prior to the $PD_{26}$-cruise make standardization almost impossible and we therefore included also comparisons of samples from the same sampling event, i.e., samples from outside the tracks vs. samples from inside the tracks (e.g. Fig. 6).

**Specific comment 11 (Page 4/5):** Conversion of biomass into carbon content; I would like to know exactly what the conversions used were, if possible. These could be included in a supplementary materials file.

**Our response:** As mentioned in the Methods section 2.2, individual macrofauna organic C values were obtained by direct measurements on an elemental analyzer, so no conversion factors were applied. Conversion factors for megafauna are presented in Supplement 1.

**Specific comment 12 (Page 5, line 20):** No details are given of the conversion used for cnidarian/ poriferan biomass to carbon content. The authors should elaborate on the use of the Tilot (1992) paper.

**Our response:** We report the conversion factors from Tilot's PhD thesis together with the other conversion factors in Supplement 2. Additionally is a digital version of the PhD thesis freely accessible online: http://archimer.ifremer.fr/doc/00000/3754/

**Specific comment 13:** The section detailing biomass to carbon content conversion would be made clearer by greater consistency in the use of the term 'biomass' to mean either the total weight of individuals, or the total carbon weight of individuals.

**Our response:** To improve the understanding of this specific section and other parts of the manuscript, we now refer to a compartment biomass as 'carbon stock' and use the term 'biomass' only for the organic carbon content of individual organisms.

**Specific comment 14 (Page 6, lines 4-9):** What literature was used to determine the coarse feedings guilds assigned to other taxa?

**Our response:** A list with references is provided in Table 2.

**Specific comment 15 (Page 6, lines 7):** It is stated that "...a further detailed classification of the macrofaunal polychaetes..." was made. However, this further detailed classification seems only to additionally subdivide deposit feeding polychaetes into surface/subsurface categories. Could this division not be made with a little effort for all invertebrate macrofauna and mega-fauna?

**Our response:** The subdivision of macrofaunal polychaetes was based on polychaete families for which detailed descriptions are available in Jumars et al. 2015. All other fauna were 'only' identified to higher taxon-level and a more specific classification in feeding types is therefore not possible/ reliable.

**Specific comment 16 (Page 7 line 23 to page 8 line 9):** Why was 'Hedge's d' used here rather than t-tests or their non-parametric equivalent? This should be clarified.

**Our response:** We used the effect size 'Hedges' d', because this is commonly used in meta studies/ analysis to compensate for the fact that we have different sample sizes for different size classes, disturbance levels and sampling events (see e.g. Koricheva, Gurevitch and Mengersen. 2013. Handbook of Meta-analysis in Ecology and Evolution. Princeton University Press).

Specific comments for the results:

**Specific comment 17 (Page 8, lines 21-23):** This comparison of the change in % biomass difference between disturbance treatments from PD0.1 to PD3 is verging on discussion. I suggest moving it to the Discussion section.

**Our response:** We report here solely data on the contributions of specific feeding types to total biomass and we therefore believe that this fits better in the Results than the Discussion.

**Specific comment 18 (Page 8, lines 24-26):** Why is 'absolute weighted Hedge's d |d+|' reported here when on page 8, lines 6-9, Hedge's d is explained in a different form? This is confusing for the reader as page 8 lines 6-9 suggests that values greater than ~0.8 represent strong effect sizes, but the results given on page 8 lines 24-26 report small values associated with the metric and the authors describe these as 'indicating a strong experimental effect'.

**Our response:**

Hedges' d is used to compare differences between carbon stocks of the same food-web compartment, e.g., megafauna deposit feeders outside vs. inside plough tracks from $PD_{0.1}$. In contrast, absolute weighted Hedges' d |d+| compares the sum of all carbon stocks from outside vs. inside plough tracks from e.g. $PD_{0.1}$. Hence, the absolute weighted Hedges' d |d+| is the summary statistic of Hedges' d. For a comparison of the effects sizes for each individual compartment (Hedges' d) and summarized over all compartments (absolute weighted Hedges' d), both types of Hedges' d are presented in tables in Supplement 3.

Specific comments for the discussion:

**Specific comment 19 (Page 9, line 28):** Suggest changing "...compared to the undisturbed sediment after 26 years" to "...compared to the undisturbed sediment 26 years after experimental mining disturbance".

**Our response:** We rephrased the sentence as follows: "the sum of all carbon flows in the food web was still significantly lower inside plough tracks compared to outside plough tracks 26 yr after experimental mining disturbance."

**Specific comment 20 (Page 10, lines 1-23):** I feel the authors are somewhat underselling the conclusions of their manuscript by placing a model limitations section so early on in their Discussion. This could be moved to later in the manuscript, perhaps to just before the conclusions.

**Our response:** We considered to move this section, but we do think that this is warranted at the beginning of the Discussion because this allows the reader to put our results that are discussed later on into perspective of the study limitations.

**Specific comment 21 (Page 11, line 7):** I am confused why the authors are discussing changes in fish respiration over long time periods (3 years) at undisturbed sites. The predictions of Jumars (1981) would be better tested by considering changes in respiration at disturbed sites very soon after the disturbance (e.g. PD0.1). I note that no fish were detected at disturbed sites at PD0.1, so simply put, this hypothesis cannot be tested with this data set.

**Our response:** We removed the discussion about respiration during $PD_3$ and concentrate on $PD_{0.1}$ as follows: "The author also predicted that the density of mobile scavengers, such as fish and lysianassid amphipods would rise shortly after the disturbance in response to the increased abundance of dying or dead organisms within the mining tracks. In fact, experiments with baits at PAP and the Porcupine Seabight (NE Atlantic) showed that the scavenging deep-sea fish *Coryphaenoides armatus* intercept bait within 30 min (Collins et al., 1999) and stayed at the food fall for $114\pm55$ min (Collins et al., 1998). Therefore, the absence of fish inside plough tracks during $PD_{0.1}$ and $PD_{0.5}$ could be related to a lack of prey in a potential predator-prey relationship (Bailey et al., 2006). However, because of the relatively small area of plough tracks (only 22% of the 10.8 $km^2$ of sediment were ploughed; Thiel et al., 1989), the low density of deep-sea fish (e.g., between 7.5 and 32 ind. $ha^{-1}$ of the dominant fish genus *Coryphaenoides* sp. at Station M; Bailey, Ruhl and Smith, 2006) and the high motility of fish, this observation is likely coincidental."

**Specific comment 22 (Page 11, line 28):** "Hence, Jumars (1981) predictions for sub-surface deposit feeders could not be tested...". Indeed, it would be easier to test Jumars' predictions if a PD0 time point was available. However, would it not be possible to test whether there is a significant difference in the density of subsurface deposit feeders at PD0.1 between the disturbance categories? Under Jumars' predictions, we would expect the density of sub-surface deposit feeders to be much reduced at 'disturbed' sites relative to 'undisturbed' sites at this time point.

**Our response:** We thank referee #2 for the suggestion how to test Jumars' (1981) predictions for subsurface deposit feeders. We used the Hedges' d and compared its development over time to investigate the recovery of subsurface deposit-feeding polychaetes: "Hence, Jumars' (1981) predictions for sub-surface deposit feeders are difficult to test, provided the natural fluctuations in PolSSDF densities that were used to calculate carbon stock. However, Hedges' d for PolSSDF was |1.47| at PD0.1 and decreased steadily to |0.66| at PD7 (Supplement 3), indicating a very strong experimental effect after the disturbance event and a constant recovery over time."

**Specific comment 23 (Page 11/12, lines 30-6):** The authors state Jumars' prediction that surface deposit feeders will be more drastically impacted by mining activities than sub-surface deposit feeders. However, they do not test this prediction, instead comparing deposit feeder ecosystem functioning to that of 'omnivores, filter- and suspension feeders and carnivores'. Please explain why. It would be possible to investigate the relative changes in surface and sub-surface deposit feeder contributions to ecosystem functioning between 'disturbed' and 'undisturbed' sites.

**Our response:** We combined the sections on surface and subsurface deposit feeders into one section also following the advice given in specific comment 22 and compare these two feeding types.

**Specific comment 24 (Page 12, lines 15-16):** "After 26 years, the relative difference in the filter and suspension feeding respiration rate was still 80%". I assume that this refers to the difference in respiration rate of filter and suspension feeders between the disturbance categories? The current text is ambiguous and could be interpreted as the difference in respiration rate between filter and suspension feeders. It is also unclear whether an 80% difference means that respiration rates at 'disturbed' sites were 80% lower than at 'undisturbed', or that respiration rates at 'disturbed' sites were 80% of those at 'undisturbed'.
**Our response:** We added the following phrase to improve the clarity of the sentence: "suspension feeding respiration rate between outside plough tracks and inside plough tracks".

**Specific comment 25 (Page 12, lines 18-19):** "...indicating a slow recovery of this feeding group". I'd argue that compared to Jumars' predictions this apparent recovery rate is relatively fast!
**Our response:** Indeed, when compared to Jumars' predictions this recovery rate is fast, however, in comparison to other feeding types, the recovery rate is rather slow.

**Specific comment 26 (Page 12, lines 23-24):** The authors complain here and elsewhere about natural variability in values making it difficult to isolate disturbance-related trends. However, the authors make no effort to identify or even simply discuss the key environmental factors which may be driving this variability.
**Our response:** See our response to general comment 5.

**Specific comment 27 (Page 12, lines 20-28):** This summary paragraph is unnecessary.
**Our response:** We removed the summary paragraph.

**Specific comment 28 (Page 13, lines 5-6):** "In contrast, filter and suspension feeders did not recover at all...". This sentence is too strongly worded. The authors state on page 12, lines 13-15, that "Directly after the initial DISCOL disturbance event, the respiration rate of filter and suspension feeders at the disturbed sediment was only 1% of the respiration rate of this feeding type at the undisturbed sediment" and on page 12, lines 15-16, that "After 26 years, the relative difference in the filter and suspension feeding respiration rate was still 80%". Whilst I agree that the respiration rate of filter- and suspension feeders is still clearly depressed at 'disturbed' sites relative to 'undisturbed', even 26 years post-disturbance, there clearly has been some recovery - perhaps even more so than might be expected! Please re-word this sentence to soften your conclusions.
**Our response:** We rephrase the sentence as follows: "In contrast, filter and suspension feeders recovered less and the relative difference in respiration rate was 79%."

**Specific comment 29 (Page 13, line 7):** The authors state that "...[ecosystem functioning] has not recovered 26 years after the experimental disturbance". However, there is clearly some evidence of recovery. Please could the authors change this statement to "...[ecosystem functioning] has not fully recovered 26 years after the experimental disturbance"?
**Our response:** We rephrased the sentence accordingly.

Specific comments for tables and figures:

**Specific comment 30 (Table 1):** Please explain what 'n' stands for. Is this the number of taxa analysed, or the number of individuals used to estimate taxon-specific biomass etc.?
**Our response:** We added the following sentence to the legend of the table: 'n' refers to the number of individuals used to estimate taxon-specific biomasses.

**Specific comment 31 (Figure 1):** There is a lot of information on this figure, and the overlap in error bars make it especially difficult to read. One option would be to plot each group separately, although this would result in a large number of graphs. Alternatively, this information may be more clearly presented as a table (as per Table 2). Why are there no error bars for the PD26 bars?
**Our response:** This point was also addressed by Peter Jumars (referee #1) and we therefore decided to present the figure without error bars which represent standard deviation and report the standard deviations together with the means in Supplement 2.

**Specific comment 32 (Figure 4):** Why are there no error bars on figure 4a? Are they simply too small to see?
**Our response:** The error bars that symbolize the standard deviations of the flows are indeed very small but were included now in Figure 4a.

Technical corrections for abstract:
**Page 1, line 27:** I, and most others, consider fish as megafauna. Please explain why they are treated separately to the other megafauna.
**Our response:** We separated fish from invertebrate megafauna because of differences in their metabolic rates. To stress that megafauna in our study only includes invertebrate megafauna, we added 'invertebrate' to megafauna throughout the manuscript.

**Page 2, line 12:** The word 'occasionally' is used twice in same sentence.
**Our response:** We removed the second 'occasionally'.

**Page 2, line 20:** Yttrium is typically considered a rare earth element.
**Our response:** We took yttrium out of the list.

**Page 2, line 25:** One could argue that there's not really such a thing as food-rich surface sediments on the deep seafloor.
**Our response:** Though the deep-sea is extremely food-limited, the surface sediments still contain more (and more labile) carbon than subsurface sediment.

**Page 3, line 2:** '10.8 km2 large circular area' – 'large' is not required.
**Our response:** We deleted 'large'.

Technical corrections for methods:
**Page 4, line 21:** 'could' should be used, not 'can'.
**Our response:** We changed the wording accordingly.

Technical corrections for the discussion:
**Page 9, line 23:** Suggest changing 'evolution' for 'change over time'.
**Our response:** We changed the wording accordingly.

**Page 9, line 27:** Put in comma after "...role of the various feeding types in the carbon cycling differs", and after "...was significantly lower".

**Our response:** We added commas accordingly.

**Page 11, line 28:** "Hence, Jumars (1981) predictions..." should be 'Hence, Jumars' (1981) predictions...'.
**Our response:** We changed it accordingly.

**Page 11, line 34:** "...deposit feeders seem to have advantages during the recovery from the DISCOL disturbance experiment...". Relative to whom?
**Our response:** To address specific comments 22 and 23, we removed this part of the text.

Technical corrections for figures and tables:
**Figure 1:** 'Figure 1' is actually referred to in the text after 'figure 2' is. Swap around the order of these figures – i.e. 'figure 2' should be renamed 'figure 1', and vice versa.
**Our response:** We renamed the figures accordingly.

**Figure 2:** Inconsistent spelling of faeces here and throughout the manuscript. Figure legend line 7 "...yellow-dashed arrow indicate..." should be '...yellow-dashed arrow indicates...'. Incorrect use of 'due to' here and throughout the manuscript. Please change to 'because of' or 'as a result of'.
**Our response:** We changed it accordingly.

**Figure 5:** Is it possible to subscript the x-axis post-disturbance times – e.g. PD0.1, for consistency with the rest of the manuscript?
**Our response:** We adjusted the figure accordingly.

[revised manuscript text omitted]